# A Myt1 family transcription factor defines neuronal fate by repressing non-neuronal genes

Joo Lee[1,2], Caitlin A Taylor[2,3], Kristopher M Barnes[4], Ao Shen[5], Emerson V Stewart[3], Allison Chen[4], Yang K Xiang[5], Zhirong Bao[4], Kang Shen[2,3]*

[1]Department of Biochemistry, Stanford University, Stanford, United States; [2]Howard Hughes Medical Institute, Stanford University, Stanford, United States; [3]Department of Biology, Stanford University, Stanford, United States; [4]Developmental Biology Program, Sloan-Kettering Institute, New York, United States; [5]Department of Pharmacology, University of California, Davis, Davis, United States

**Abstract** Cellular differentiation requires both activation of target cell transcriptional programs and repression of non-target cell programs. The Myt1 family of zinc finger transcription factors contributes to fibroblast to neuron reprogramming in vitro. Here, we show that *ztf-11* (*Zinc-finger Transcription Factor-11*), the sole *Caenorhabditis elegans* Myt1 homolog, is required for neurogenesis in multiple neuronal lineages from previously differentiated epithelial cells, including a neuron generated by a developmental epithelial-to-neuronal transdifferentiation event. *ztf-11* is exclusively expressed in all neuronal precursors with remarkable specificity at single-cell resolution. Loss of *ztf-11* leads to upregulation of non-neuronal genes and reduced neurogenesis. Ectopic expression of *ztf-11* in epidermal lineages is sufficient to produce additional neurons. ZTF-11 functions together with the MuvB corepressor complex to suppress the activation of non-neuronal genes in neurons. These results dovetail with the ability of Myt1l (Myt1-like) to drive neuronal transdifferentiation in vitro in vertebrate systems. Together, we identified an evolutionarily conserved mechanism to specify neuronal cell fate by repressing non-neuronal genes.
DOI: https://doi.org/10.7554/eLife.46703.001

*For correspondence:
kangshen@stanford.edu

## Introduction

Transcriptional repressors such as RE1-silencing transcription factor (REST) and Hairy/Enhancer of Split (Hes) repress neuronal genes in non-neuronal cells (*Ballas et al., 2005*; *Chen et al., 1998*; *Chong et al., 1995*; *Grill et al., 2012*; *Ishibashi et al., 1995*; *Ohsako et al., 1994*; *Schoenherr and Anderson, 1995*). However, it is unknown whether transcriptional repressors of non-neuronal genes are required in neuronal precursors to specify neuronal fate during development. The Myt1 family of C2HC-type zinc finger transcription factors contributes to fibroblast to neuron reprogramming in vitro by repressing Notch signaling (*Bellefroid et al., 1996*; *Mall et al., 2017*; *Vasconcelos et al., 2016*; *Vierbuchen et al., 2010*). The Myt1 family factors were first shown to regulate neurogenesis in *Xenopus* gastrula embryos, where X-MyT1 is expressed in neuronal precursors along with classical proneural genes (*Bellefroid et al., 1996*). Mammalian Myt1 family proteins, Myt1, Myt1l, and St18, are also highly expressed in developing nervous systems and are required for proper migration of neuronal precursors into the subventricular zone and cortical plate (*Mall et al., 2017*; *Vasconcelos et al., 2016*). Myt1 transcriptionally represses Notch signaling, primarily by repressing the transcription factor Hes1, which inhibits neuronal cell fate (*Mall et al., 2017*; *Vasconcelos et al., 2016*). The ability of Notch intracellular domain to repress neurogenesis is neutralized by overexpression of Myt1 family proteins (*Bellefroid et al., 1996*; *Mall et al., 2017*). Based on these results,

**eLife digest** The human body contains many cell types that each have different job and can look very different from each other. However, each of the cells in an individual's body contains almost exactly the same genes, because all of them share the same DNA inherited from the individual's parents. Cells therefore become different from one another by controlling the activity of sets of genes. They do this by using proteins called transcription factors, which find specific genes and turn them either on or off.

Nerve cells or neurons form and develop in a process called neurogenesis. During neurogenesis, some genes including those specific to neurons need to be switched on while other non-neuronal genes need to be switched off. The "off-switch" is particularly important when neurons are generated by conversion from skin cells, which sometimes happens in animals. Before these cells can become mature nerve cells, they require transcription factors to ensure that skin-specific genes are off.

The transcription factors turning on nerve cell-specific genes are well-understood, but far less is known about those that turn off other genes. Lee et al. therefore set out to search for transcription factors that might switch off non-neuronal genes during neurogenesis and focused on one transcription factor that is known to be important for the development of nerve cells in a variety of animal species.

Experiments using the worm *C. elegans* revealed that this transcription factor – called ZTF-11 in worms – was present in all cells destined to be nerve cells, but not in cells that would assume other roles. These experiments are possible with *C. elegans* because the final role, or 'fate', of each cell in the body are already known, all the way from the fertilized egg to the adult.

Further work, using genetically engineered worms revealed that ZTF-11 worked by turning off genes that are related to the development of non-nerve cells. Deleting the gene for ZTF-11 in immature nerve cells allowed these cells to turn on different sets of genes and resulted in adult worms with fewer mature nerve cells than normal worms. On the other hand, forcing other cell types (which would not normally become part of the nervous system) to produce ZTF-11 was sufficient to convert them into nerve cells.

These results are an important step forward in understanding how nerve cells are built in the developing body, especially how nerve cells can be made from other cell types. In the future, this knowledge could be used to help people with diseases of the nervous system, such as Parkinson's disease.

DOI: https://doi.org/10.7554/eLife.46703.002

it has been proposed that Myt1 family proteins counteract lateral inhibition and subsequently commit neuronal progenitors to terminal differentiation.

Recent in vitro studies showed that Myt1l, together with the proneural gene Ascl1 and the neuronal transcription factor Brn2, are sufficient to induce transdifferentiation (TD) into neurons from various cell types (*Masserdotti et al., 2016*; *Vierbuchen et al., 2010*; *Wapinski et al., 2013*). Interestingly, a number of non-neuronal mouse embryonic fibroblast (MEF) signature genes were also found to be repressed by Myt1l during neuronal transdifferentiation. Furthermore, co-expression of Myt1l reduced efficiency of MyoD-induced myocyte differentiation in vitro (*Mall et al., 2017*). Consistent with a role for Mytl1 in transcriptional repression during neuronal transdifferentiation, Myt1l was found to be associated with transcriptional corepressor complexes, including the Sin3 histone deacetylase complex (Sin3-HDAC), to mediate repression of non-neuronal genes (*Romm et al., 2005*). Redundancy between Myt1 family proteins has prevented mouse models from providing insight into the developmental functions of Myt1 (*Wang et al., 2007*). As a result, the in vivo functions of Myt1 family proteins during development remain poorly understood.

In *C. elegans*, ZTF-11 is the sole Myt1 family homolog containing the characteristic C2HC zinc finger domains. The DNA-binding zinc finger domains of ZTF-11 exhibit a high degree of conservation in amino acid sequence compared to other Myt1 family members (*Figure 1—figure supplement 1*). Both ZTF-11 and vertebrate Myt1 family proteins recognize the same consensus sequence (AAGTT) in vitro (*Mall et al., 2017*; *Narasimhan et al., 2015*; *Vasconcelos et al., 2016*). Apart from the zinc

finger domains, ZTF-11 and other Myt1 family proteins are poorly conserved in sequence, including regions that interact with the Sin-3-HDAC complex (*Mall et al., 2017*; *Romm et al., 2005*) (*Figure 1—figure supplement 1*).

Here, we demonstrate that a Myt1 family protein is required for in vivo developmental neurogenesis of specific lineages as well as transdifferentiation by characterizing the in vivo functions of *ztf-11*. We found that ZTF-11 is expressed exclusively in neuronal precursors at single-cell resolution during embryonic and postembryonic neurogenesis. Remarkably, ZTF-11 is required for epithelial-to-neuronal transdifferentiation during *C. elegans* development, suggesting that in vivo transdifferentiation utilizes genetic programs similar to those required for neuronal reprogramming in vitro. We also found that ZTF-11 is necessary and sufficient for postembryonic neurogenesis from a non-neuronal precursor. In these lineages, we show that *ztf-11* represses expression of non-neuronal genes to allow establishment of neuronal identity. Unexpectedly, *ztf-11* does not function as a repressor of the Hes1 ortholog *lin-22* in this context. Instead, our genetic data support the model that ZTF-11 acts downstream of Hes1 to promote neuronal differentiation. We further show that ZTF-11 mediates transcriptional repression through directly binding to MuvB co-repressor complex, but not the Sin3-HDAC complex. Taken together, our results indicate that neurogenesis requires repression of non-neuronal programs by Myt1 family proteins in addition to activation of neuronal programs.

## Results

### Myt1 family homolog ZTF-11 is expressed in neural precursors

To investigate the role of *ztf-11* in development, we first examined the expression pattern of ZTF-11 by endogenously tagging *ztf-11* with a C-terminal GFP via CRISPR/Cas9 genome editing. To facilitate *Cre* recombinase-mediated conditional knock-outs, we used the same approach to insert two loxP sites in the first intron and 3'UTR of *ztf-11::gfp* (*Figure 1A*). Insertion of GFP and loxP sites in the *ztf-11* locus did not yield any overt phenotypes. In comparison, a deletion spanning one of two zinc-finger domains (*tm2315*), which likely abolished DNA binding, was homozygous lethal and produced severely paralyzed, developmentally arrested L1 larvae when maintained with the hT2 balancer chromosome (data not shown). As expected, we found that ZTF-11::GFP fluorescence was predominantly localized to nuclei (*Figure 1B*).

Myt1 family transcription factors have been shown to be expressed early in neural precursors in the neural plate of *Xenopus* gastrula embryos and the developing CNS of rat embryos (*Bellefroid et al., 1996*; *Kim et al., 1997*). As with vertebrate orthologs, we found that ZTF-11::GFP is expressed in neural precursors starting in the mid-gastrula embryo (~100-cell stage) (*Figure 1B* and *Video 1*). ZTF-11::GFP expression was strongest during the late gastrula to lima-bean embryonic stages, coinciding with the birth of most embryonically generated neurons. ZTF-11::GFP expression became weaker in subsequent stages of embryogenesis. Around the time of hatching, ZTF-11::GFP could only be detected in a small number of neuronal nuclei. Post-embryonic ZTF-11::GFP expression showed a similar pattern; ZTF-11::GFP expression was transiently observed in postembryonic neuroectoblasts, such as Pn, Q, and V5 cells, but was quickly extinguished in postmitotic neurons (*Figure 1—figure supplement 2*). Together, these data suggest that ZTF-11 is transiently expressed in neuronal precursors and postmitotic neurons.

We next performed embryonic lineage tracing with 4-D microscopy to further characterize *ztf-11* expression with single-cell resolution. The invariant cell division patterns during embryogenesis made it possible to reliably track cell lineage with a nuclear marker (*Bao et al., 2006*; *Sulston et al., 1983*). Examining ZTF-11::GFP fluorescence throughout embryonic development, we found that *ztf-11* is expressed in the vast majority of lineages that generate neurons but is rarely expressed in lineages that do not produce neurons (*Figure 1C*, also see *Supplementary file 1* for full lineage diagram). At the 350-cell stage, 126 of 145 (87%) neuronal precursor cells expressed ZTF-11::GFP. All six neuroectoblasts (P7/8, P5/6, and P3/4) that did not express ZTF-11 at the 350-cell stage showed postembryonic expression of ZTF-11. In contrast, only 15 of 195 (7%) non-neuronal precursor cells expressed ZTF-11::GFP. In *C. elegans*, the majority of neurons are generated from the neuroectodermal AB lineage, while a small number of neurons are produced by other lineages (*Sulston et al., 1983*). Strong correlations between *ztf-11* expression and neuronal cell fate were evident in all

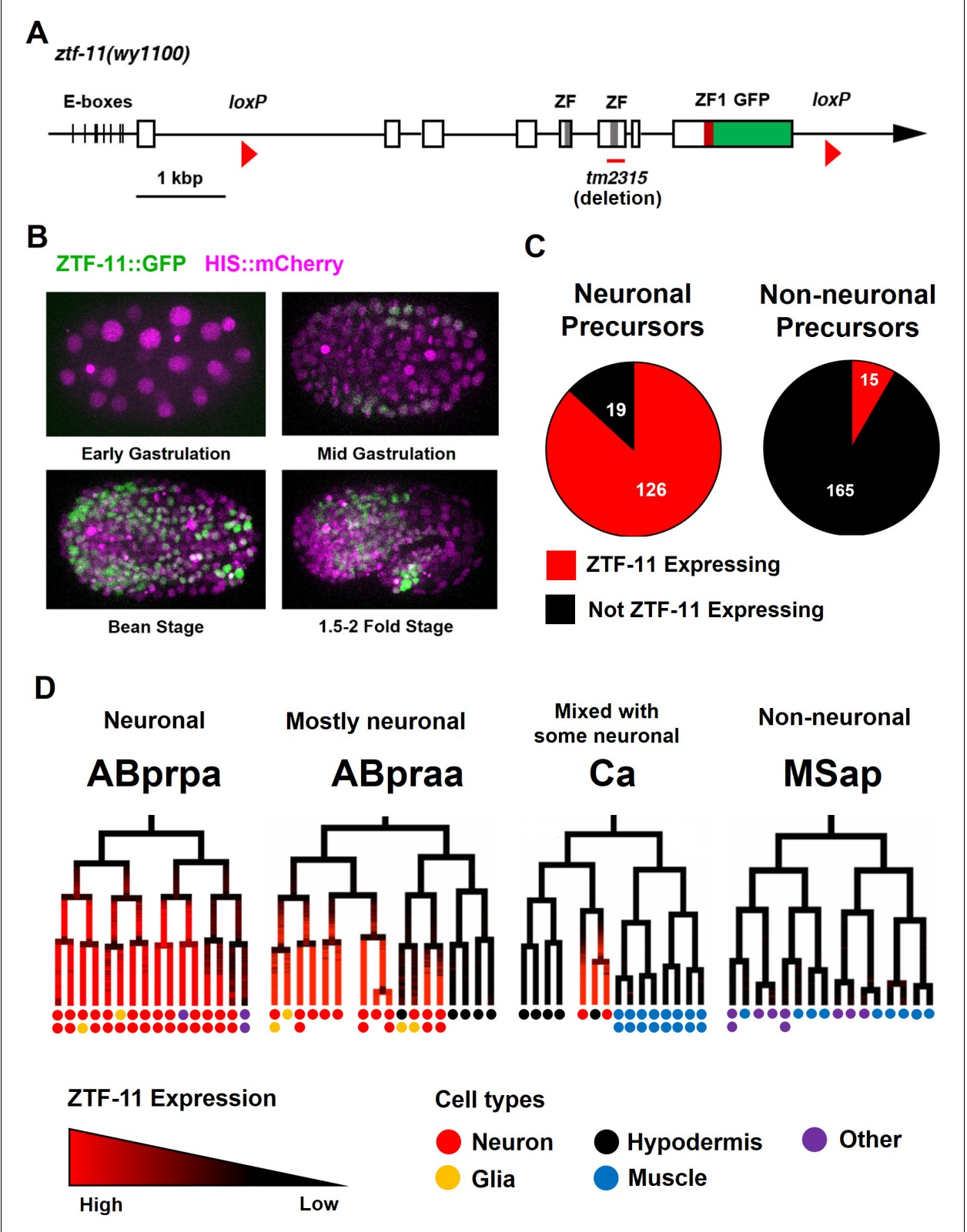

**Figure 1.** Myt1 family homolog ZTF-11 is expressed in neuronal precursors. (A) Schematic showing *ztf-11::gfp (wy1100)*. ZF, C2HC zinc-finger domain. ZF1, ZF1 zinc-finger domain. Magenta line underneath the second ZF denotes the area deleted in *tm2315*. See also *Figure 1—figure supplement 1* for evolutionary conservation of Myt1 family proteins. (B) ZTF-11::GFP expression in neuronal precursor populations during embryogenesis. Also see *Video 1*. (C) Quantification of ZTF-11::GFP expressing cells in neuronal precursors and non-neuronal precursors at 350 cell stage. A neuronal precursor

*Figure 1 continued on next page*

*Figure 1 continued*

was defined by any cells giving rise to non-pharyngeal neurons. (D) Selected lineage diagrams showing correlation between ZTF-11::GFP expression and terminal cell fates in selected sub-lineages. Each dot under the line represents the ultimate cellular fate. In many cases, cells undergo additional round of cell division past automated lineage tracing and result in two daughter cells (indicated by two dots under each line). See also *Supplementary file 1* for full lineage diagram with ZTF-11::GFP expression and their terminal fates.

DOI: https://doi.org/10.7554/eLife.46703.003

The following figure supplements are available for figure 1:

**Figure supplement 1.** Myt1 family proteins share conserved C2HC zinc-finger domains.
DOI: https://doi.org/10.7554/eLife.46703.004
**Figure supplement 2.** ZTF-11 is expressed in postembryonic neuroectoblasts.
DOI: https://doi.org/10.7554/eLife.46703.005

branches of the lineages at single-cell resolution (*Figure 1D*). These observations suggest that ZTF-11 plays a broad role in neurogenesis.

## ZTF-11 is a direct transcriptional target of proneural bHLH genes

The expression pattern of the Myt1 family proteins closely follows proneural bHLH (*basic helix-loop-helix*) genes in vertebrate systems (*Bellefroid et al., 1996*; *Kim et al., 1997*). The vertebrate Myt1 family proteins are direct transcriptional targets of proneural genes, including Ascl1 (*Mall et al., 2017*; *Vasconcelos et al., 2016*; *Wapinski et al., 2013*). *C. elegans* proneural genes are conserved through evolution and act as master regulators of neurogenesis. These proneural genes are expressed in neuronal precursors and differentiating neurons (*Frank et al., 2003*; *Hallam et al., 2000*; *Murray et al., 2012*; *Zhao and Emmons, 1995*). To investigate whether neural precursor-specific expression of ZTF-11 was directly controlled by proneural genes in *C. elegans*, we first asked whether proneural genes are required for ZTF-11 expression.

All *C. elegans* proneural genes, including HLH-3/Achaete-Scute, LIN-32/Atonal, NGN-1/neurogenin, and CND-1/NeuroD, form heterodimers with HLH-2/Daughterless and bind to canonical E-boxes (CANNTG) to regulate transcription of target genes (*Grove et al., 2009*). We tested whether *hlh-2* functions with proneural genes to regulate *ztf-11* expression. Since *hlh-2* is linked to *ztf-11*, we constructed a *ztf-11* transcriptional reporter fusion by placing the *ztf-11* promoter upstream of Histone:: GFP (HIS::GFP) and confirmed that the transcriptional reporter reproduces the endogenous expression pattern of ZTF-11 (*Figure 2—figure supplement 1*). We found that hypomorphic *hlh-2(tm1768)* mutant showed a strong reduction of *ztf-11* transcriptional reporter signal (78%) in comparison to wild type (*Figure 2A*), suggesting that *hlh-2* is required for proper expression of *ztf-11*. In contrast to the essential *hlh-2*, the proneural dimer partners of HLH-2 are redundantly expressed at early stages of neuronal development (*Grove et al., 2009*; *Murray et al., 2012*). Consistent with this redundancy, endogenous ZTF-11::GFP expression was largely unperturbed in single mutants of *hlh-3 (ot354)*, *cnd-1(ju29)* or *ngn-1(ok2200)* (data not shown). We found that *lin-32(n372)* mutant showed

loss of ZTF-11 expression in the postembryonic postdeirid lineage, where LIN-32 functions to generate sensory neurons (Figure 8A).

Furthermore, we identified multiple canonical E-box sequences upstream of the *ztf-11* coding region (*Figures 1A* and *2B*). Mutating these E-box sequences (CANNTG to *ACNNAG*) (Δ*E-box*) caused a severe reduction (79%) of Δ*E-box* reporter signal compared to wild-type reporter (*Figure 2B*). These results are consistent with findings in vertebrates and suggest that proneural genes and HLH-2 together activate the expression of *ztf-11* in neuronal precursors through the E-boxes in the *ztf-11* promoter.

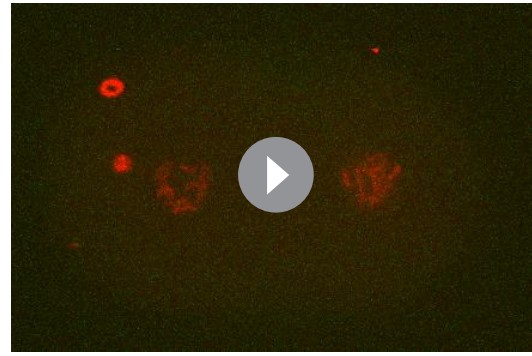

**Video 1.** ZTF-11 expression in developing *C. elegans* embryo.
DOI: https://doi.org/10.7554/eLife.46703.006

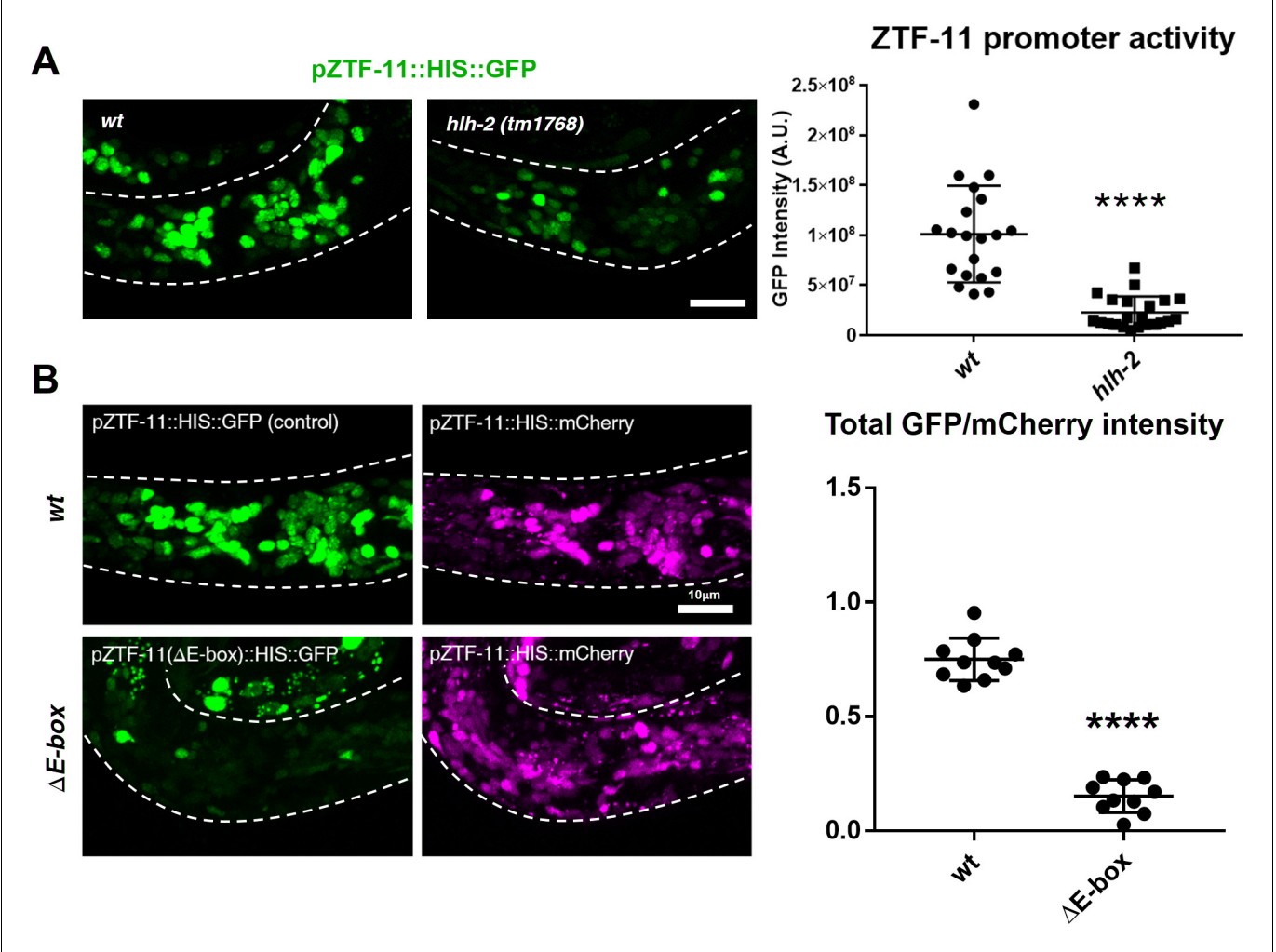

**Figure 2.** ZTF-11 is a direct transcriptional target of proneural bHLH genes. (**A**) *ztf-11* transcriptional reporter expression in head neurons is strongly decreased in E/daughterless homolog *hlh-2(tm1768)* hypomorphic mutant. Left, representative confocal images of *ztf-11* transcriptional reporter in *wt* or *hlh-2*. Synchronized early L1 animals were used in the experiment. Right, transcriptional activity was measured in total GFP intensity within each animal's head (white dashed outlines), n = 20 and 22, respectively. Error bars, S.D. ****p<0.0005, Student's t-test two-tailed. See *Figure 2—source data 1* for numerical data. (**B**) E-boxes are required for ZTF-11 transcriptional reporter expression. wt, wild-type *ztf-11* promoter driving GFP and mCherry reporters. ΔE-box, 8 E-box sequences (CANNTG) nearest to *ztf-11* tss were mutated to *ACnnAG* for GFP driving promoter, mCherry expression is under wild-type *ztf-11* promoter as expression level control. Synchronized early L1 animals were used in the experiment. Left, representative confocal images. Right, relative intensities between GFP and mCherry within each animal's head (white dashed outlines) were quantified. n = 10 each. Error bars, S.E.M. **p<0.05, Student's t-test two-tailed. See *Figure 2—source data 2* for numerical data.

DOI: https://doi.org/10.7554/eLife.46703.007

The following source data and figure supplement are available for figure 2:

**Source data 1.** Raw fluorescence intensity values of ZTF-11 transcriptional reporter in wild-type or *hlh-2(tm1768)* animals.

DOI: https://doi.org/10.7554/eLife.46703.009

**Source data 2.** Normalized fluorescence intensity values of wild-type or ΔE-box transcriptional reporters.

DOI: https://doi.org/10.7554/eLife.46703.010

**Figure supplement 1.** Expression pattern of ZTF-11 transcriptional reporter.

DOI: https://doi.org/10.7554/eLife.46703.008

## ZTF-11 is required for epithelial-to-neuronal transdifferentiation in vivo

We next investigated the requirement of *ztf-11* for neuronal fate determination in three contexts: in vivo transdifferentiation, postembryonic neurogenesis, and embryonic neurogenesis. An epithelial-to-neuronal transdifferentiation event occurs invariantly during normal *C. elegans* development

(*Jarriault et al., 2008*), providing a model for investigating genetic pathways involved in neuronal transdifferentiation in vivo. The rectal epithelial Y cell undergoes an epithelial-to-neuronal transdifferentiation event to form the motor neuron PDA in a stepwise process; epithelial identity is first lost and neuronal identity is subsequently acquired (*Jarriault et al., 2008*; *White et al., 1986*) (*Figure 3A*). At the L1 stage, the Y cell is part of the rectal epithelium. During the L2 stage, the Y cell gradually loses its epithelial fate (Y.0) while migrating anteriorly and gaining neuronal markers (Y.1). In the L3 stage, the Y cell becomes the PDA neuron and extends an axon while the P12.pa cell replaces the Y cell in the anal epithelium (*Jarriault et al., 2008*; *Sulston and Horvitz, 1977*; *White et al., 1986*) (*Figure 3A–B*).

We first asked whether ZTF-11 functions during the Y-PDA transdifferentiation event. Using the endogenously-tagged ZTF-11::GFP, we found that *ztf-11* was expressed in the Y cell in early L2 animals at the start of the transdifferentiation process (*Figure 3C*). ZTF-11 expression coincided with the initial withdrawal of the Y cell from the rectum, suggesting that ZTF-11 may mediate the early dedifferentiation step of transdifferentiation (Y.0). We next asked if *ztf-11* is required for transdifferentiation by generating a conditional deletion strain, in which we used *egl-26*::Cre to delete *ztf-11* from the Y cell in the postembryonic lineage. Conditional deletion of *ztf-11* in the Y cell led to the loss of PDA neuronal markers, including EXP-1 and COG-1, suggesting that PDA was not generated (*Figure 3D,F*, and *Figure 3—figure supplement 1*).

We next asked whether the transdifferentiation defect is due to a failure to eliminate epithelial identity or to acquire neuronal identity. We found that epithelial markers in the Y cell (EGL-26 and COL-34) persisted throughout development in *ztf-11 cKO* animals (*Figure 3E* and *Figure 3—figure supplement 1*). Moreover, the persistent Y cell in *ztf-11 cKO* animals retained its original rectal niche location and morphology during development (*Figure 3E–F*). These results argue that *ztf-11* functions to eliminate epithelial identity in the Y cell and allows for subsequent acquisition of neuronal PDA identity. This results are in accordance with Myt1l's function in in vitro neuronal transdifferentiation (*Mall et al., 2017*; *Vierbuchen et al., 2010*). While mammalian models of in vivo neuronal transdifferentiation have not yet been described, Myt1 family factors may function as key evolutionarily conserved repressive factors in transdifferentiation events.

## ZTF-11 is required for loss of epithelial identity and subsequent neuronal differentiation in sensory organ development

Neurogenesis via developmental transdifferentiation is rare in the animal kingdom. Most neurons are generated through asymmetric cell divisions and quickly adopt a neuronal cell fate after mitosis. We next assessed *ztf-11*'s function in this more common cellular pathway of neurogenesis by studying a postembryonic neuronal lineage. The *C. elegans* postdeirid is a simple sensory organ comprised of two morphologically and functionally distinct sensory neurons, PVD and PDE, and a pair of sheath (PDEsh) and socket (PDEso) glia that support the PDE sensory dendrite (*White et al., 1986*). These four cells are born postembryonically from the V5 seam cell, which forms part of the lateral epidermis (*Figure 4A–B* and *Figure 4—figure supplement 2A–B*). Unlike V5 or tail T lineages, parallel lateral epidermal seam cells of V lineages (V1-4 and V6) do not give rise to any neural progeny (*Sulston and Horvitz, 1977*). Previous genetic studies have identified mutations in *lin-32,* the homolog of *Drosophila atonal* and mammalian Atoh1, which cause the V5 lineage to lose postdeirid neuroblast cell fate and instead to adopt a V1-4-like cell fate, establishing LIN-32 as the master regulator of postdeirid development (*Kenyon, 1986*; *Zhao and Emmons, 1995*). It is interesting to note that homologs of the *atonal* family of proneural bHLH factors function in the development of various mechanosensory modalities, including mammalian inner ear hair cells (*Bermingham et al., 1999*), *Drosophila* chordotonal organs (*Jarman et al., 1993*), and the *C. elegans* postdeirid and male sensory rays (*Zhao and Emmons, 1995*), suggesting that atonal family bHLH genes drive a conserved genetic program.

To investigate the expression pattern of ZTF-11 in V5 postdeirid lineage in detail, we followed the V5 lineage using two fluorescent markers, endogenously labeled ZTF-11::GFP and HIS:mCherry driven by the *lin-32* promoter. Starting from mid L1, ZTF-11::GFP was observed in the posterior daughter cell of V5, which gives rise to the neurons, but not in V1-4 nor in the anterior daughter of V5, which generate epidermal cells (hyp7) and seam cells. Within the V5 lineage, ZTF-11::GFP was maintained in neurons and glia but turned off in the non-neuronal precursors (*Figure 4A–B* and *Figure 4—figure supplement 1C*). Unexpectedly, the *lin-32* transcriptional reporter showed dynamic

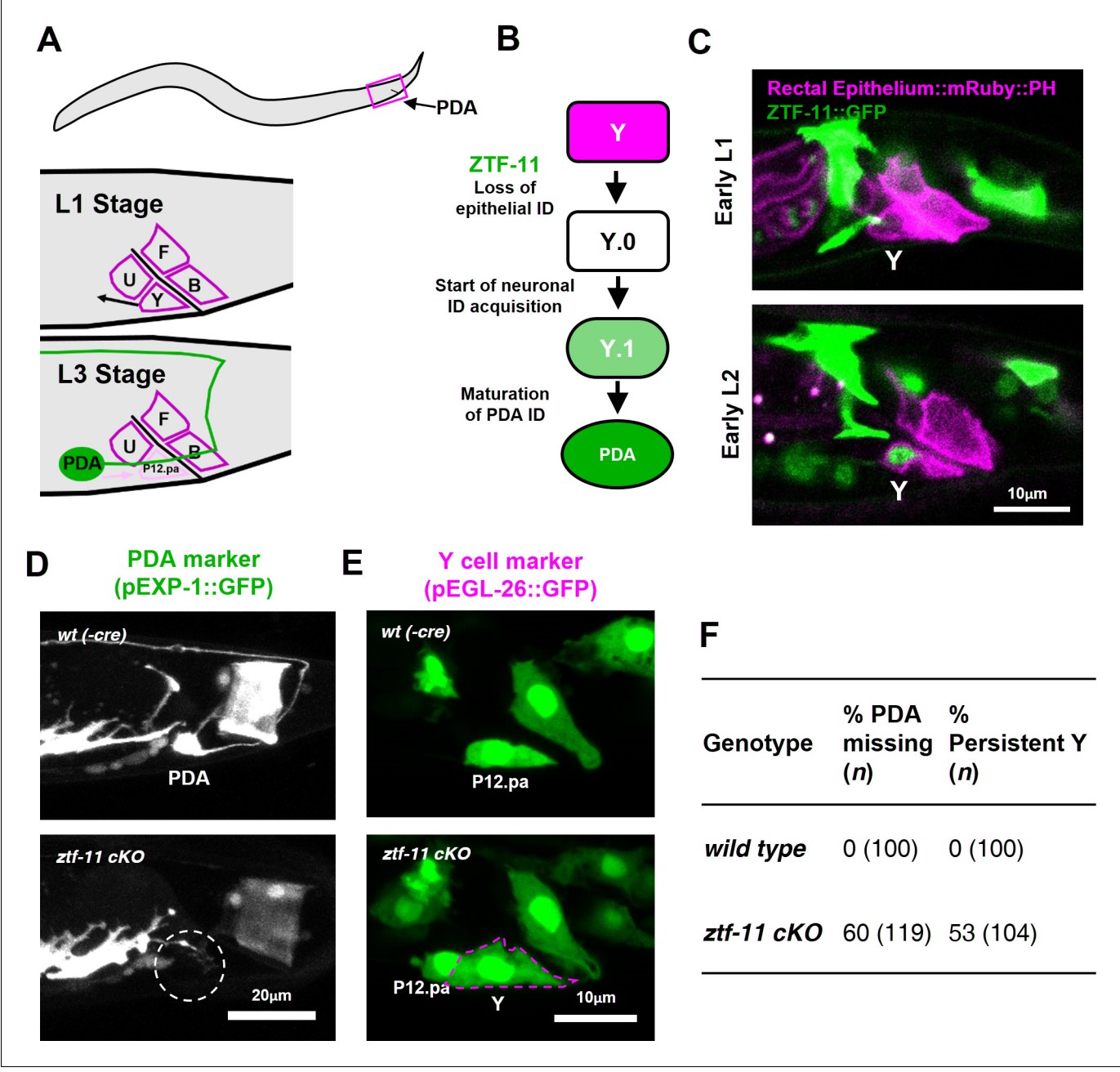

**Figure 3.** ZTF-11 is required for epithelial-to-neuronal transdifferentiation in vivo. (A) PDA neuron, located in preanal ganglion, arises from stereotyped transdifferentiation of rectal epithelial cell Y. (B) Schematic diagram of Y-to-PDA transdifferentiation event. Rectal epithelium Y cell withdraws from its rectal niche starting late L1 and loses epithelial markers to yield an intermediate cell (Y.0). During L2, Y gains neuronal markers and morphology (Y.1) to become mature PDA in L3. Meanwhile, P12.pa cell take place of Y in the rectal epithelium. Letter color code; magenta (epithelial), green (neuronal) (C) Representative images of ZTF-11::GFP expression during Y-PDA transdifferentiation. Rectal epithelium was labeled with *egl-26* marker (magenta, rectal epithelium::mRuby::PH). (D) PDA marker, *exp-1*, expression in wild type (*wt(-Cre)*) or *ztf-11* conditional knock-out (*ztf-11 cKO*). Dashed circle indicates the position of PDA cell body. (E) Y cell marker, *egl-26*, expression in wild type (*wt(-Cre)*) or *ztf-11* conditional knock-out (*ztf-11 cKO*). Magenta dashed line outlines the retained Y cell. See also **Figure 3—figure supplement 1** for exclusivity between PDA marker (*cog-1*) and rectal epithelium marker (*col-34*). (F) Quantification of PDA marker (*exp-1*) loss (from D) and Y marker retention (from E) phenotypes.

DOI: https://doi.org/10.7554/eLife.46703.011

The following figure supplement is available for figure 3:

**Figure supplement 1.** Rectal epithelial and neuronal fates are mutually exclusive during transdifferentiation.
DOI: https://doi.org/10.7554/eLife.46703.012

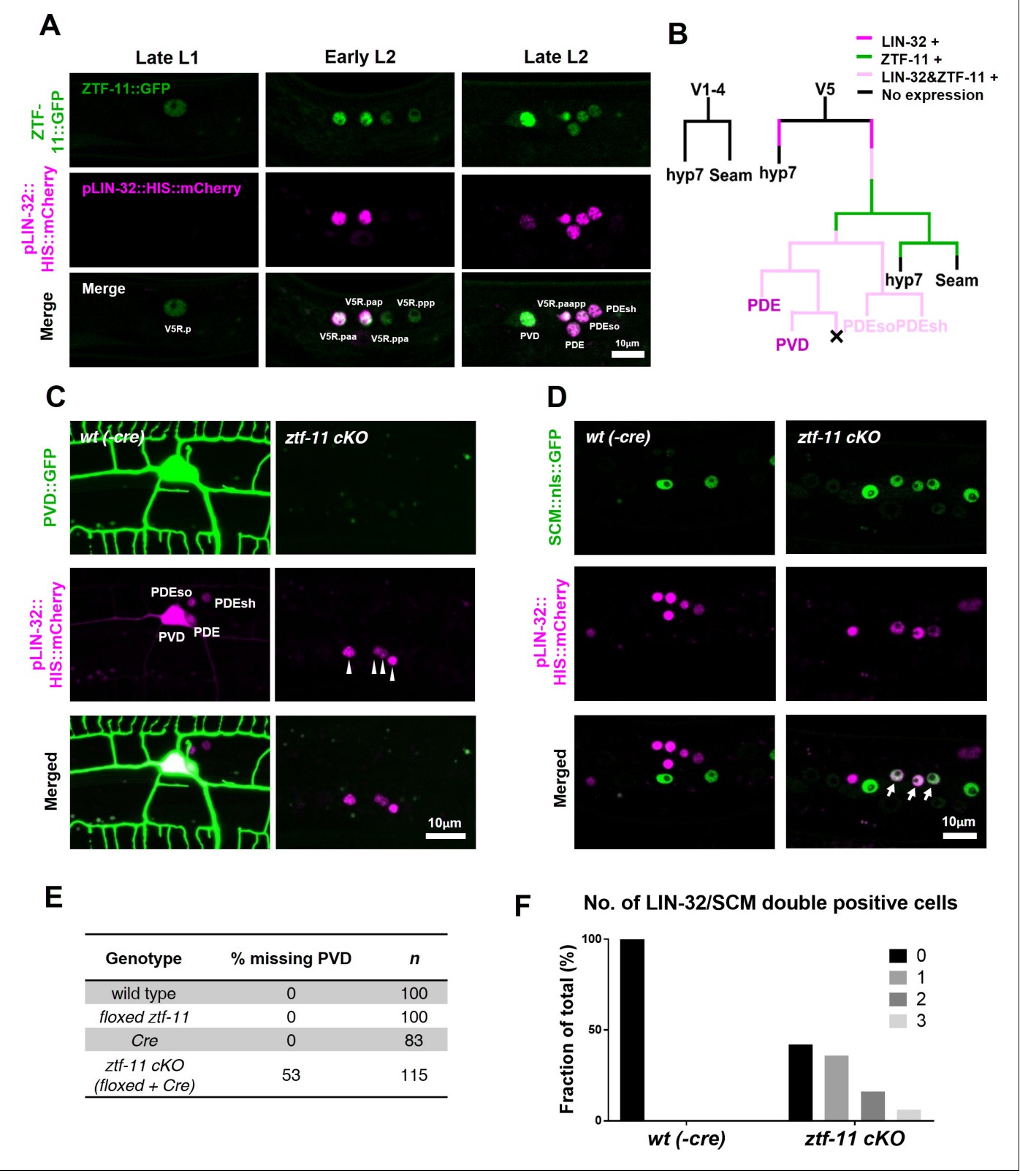

**Figure 4.** ZTF-11 is required for loss of epithelial identity and subsequent neuronal differentiation in sensory organ development. (**A**) *ztf-11* and *lin-32* are expressed during postdeirid development. Both genes are expressed throughout postdeirid neuroblast (V5.pa) divisions to yield two neuron and two glia. See also *Figure 4—figure supplement 1* for full sequence of *ztf-11* and *lin-32* expression during postdeirid development. (**B**) V lineage diagram showing cell divisions resulting in postdeirid neurogenesis during L2 larval development. In contrast to V1-4, V5 dynamically expresses ZTF-11

*Figure 4 continued on next page*

*Figure 4 continued*

and LIN-32 during neurogenesis. V5 and other seams cells undergo asymmetric cell divisions where the posterior daughters remain as seam cells and the anterior daughters join the epidermal syncytium (hyp7). Line colors denote expression of *lin-32* or *ztf-11*. Letter color code; black (epithelial), magenta (neuronal), pink (glial). (C) PVD marker, *ser-2*, expression in wild type (*wt(-Cre)*) or *ztf-11* conditional knock-out (*ztf-11 cKO*). Note that *lin-32*-expressing nuclei are still present, suggesting that postdeirid cells are born, but fails to adopt neuronal fate. (D) Seam cell marker, *scm* (wIs78), expression in wild-type expression in wild type (*wt(-Cre)*) or *ztf-11* conditional knock-out (*ztf-11 cKO*). For C and D, postdeirid cells are labeled with *lin-32* marker. (E) Quantification of missing PVD phenotype from C. (F) Quantification of seam cell fate retention phenotype. *n* = 50 for each genotype. See *Figure 4—source data 1* for numerical data.

DOI: https://doi.org/10.7554/eLife.46703.013

The following source data and figure supplements are available for figure 4:

**Source data 1.** Number of cell nuclei expressing both LIN-32 and SCM fate markers.

DOI: https://doi.org/10.7554/eLife.46703.016

**Source data 2.** Raw fluorescence intensity values of floxed ZTF-11::GFP in wild-type(-cre) or *ztf-11 cKO* animals.

DOI: https://doi.org/10.7554/eLife.46703.017

**Figure supplement 1.** ZTF-11 functions in an epithelial precursor to promote development of a simple sensory organ.

DOI: https://doi.org/10.7554/eLife.46703.014

**Figure supplement 2.** Neuronal and glial cell fates are lost in *ztf-11* conditional knock-out.

DOI: https://doi.org/10.7554/eLife.46703.015

expression in the V5 lineage. *lin-32* expression was detected prior to expression of ZTF-11::GFP. However, *lin-32* could not be detected in late L1, while ZTF-11 expression was maintained throughout. *lin-32* reappeared again in the postdeirid neuroblast V5.pa, but not in epithelial sister V5.pp (*Figure 4B* and *Figure 4—figure supplement 1C*). While the significance of the *lin-32* expression dynamics remains unclear, both LIN-32 and subsequent ZTF-11 expression were correlated with neuronal and glial cell fate.

To investigate the role of ZTF-11 in postdeirid neurogenesis, we generated a seam cell-specific *ztf-11* conditional knock-out (cKO) by expressing Cre recombinase under the seam-cell-specific *nhr-81* promoter to excise the ZTF-11::GFP locus. To determine efficiency of the *ztf-11* cKO, we measured ZTF-11::GFP intensity in the postdeirid lineage. We found near complete loss of ZTF-11::GFP expression in 70% of cKO animals. However, 30% of cKO animals showed only partial knock-down that fell within the wild type range of ZTF-11 expression, likely due to perdurance of *ztf-11* mRNA or protein (*Figure 4—figure supplement 1A–B*). Partial penetrance observed in subsequent phenotypic analyses of *ztf-11* cKO was most likely attributable to these limitations of the cKO approach.

We first scored neuronal reporters to examine whether the postdeirid neurons, PVD and PDE, could adopt a neuronal fate in the absence of ZTF-11. We found that approximately 50% of PVD and PDE neurons had lost their neuronal fate, as reflected by loss of the respective cell-type-specific markers, *ser-2* and *dat-1*, as well as loss of the pan-neuronal *rab-3* marker (*Figure 4C and E*, and *Figure 4—figure supplement 1C–E*). Additionally, we found a similar loss of glial markers from the socket and sheath glia that function with the PDE neuron (*Figure 4—figure supplement 1F*), suggesting that ZTF-11 is also required for glial fate in the postdeirid lineage. The number of LIN-32-expressing V5 lineage cells was unchanged in *ztf-11 cKO* animals (*Figure 4C–D*), indicating that V5 lineage cells still undergo the stereotyped cell divisions that would generate neurons and glia in wild-type animals. This is in contrast to *lin-32* mutants, which do not go through postdeirid cell divisions, and instead exclusively adopt epithelial V1-4-like lineages (*Kenyon, 1986*; *Zhao and Emmons, 1995*).

Since the neurons and glia of the postdeirid originate from an epithelial precursor, proper differentiation into their terminal fate likely requires both the loss of epithelial identity and the acquisition of neuronal/glial identities, similar to transdifferentiation. In wild-type animals, the expression of the seam cell fate marker SCM::GFP is invariably lost in *lin-32*-positive postdeirid cells as they acquire neuronal/glial fate. Strikingly, in *ztf-11* cKO animals, we observed that some *lin-32*-positive postdeirid cells retained seam cell fate marker expression (*Figure 4D and F*), suggesting that *ztf-11* was required for the removal of epithelial identity preceding the acquisition of neuronal identity. These data are consistent with the notion that ZTF-11 plays a role in eliminating epithelial fate in differentiating V5 lineage cells.

## ZTF-11 is sufficient to generate neurons from epithelial cells by repressing epithelial identity

We next asked whether *ztf-11* was sufficient to produce neurons. We ectopically expressed ZTF-11 in non-neurogenic V1-4 seam cells where *ztf-11* is not normally expressed. Remarkably, we found that ectopic expression of ZTF-11 led to transformation into a neuronal lineage. In 45% of transgenic animals expressing ZTF-11 in seam cells, we found additional cells expressing the PVD cell marker anterior to the wild-type PVD (*Figure 5A*). In addition, ectopic PVD-like cells showed the characteristic 'dendritic menorah' morphology of PVD neurons (*Albeg et al., 2011*). The positions of the ectopic PVDs were consistent with positions of V1-4 seam cell precursors (*Figure 5A–B*). Similarly, additional PDE-like cells were identified based on the presence of the PDE cell marker and PDE morphology (*Figure 5C–D*). In contrast to the proneural activity of ZTF-11, additional glia-like cells could not be identified (*Figure 5D*), suggesting additional requirements for glia development.

Our genetic data suggested that ZTF-11 is required to eliminate epithelial identity in developing neurons and glial cells. We next asked whether *ztf-11* is sufficient to eliminate epithelial identity by ectopically expressing ZTF-11 in seam cell lineages. In animals ectopically expressing ZTF-11 in seam cells, we indeed found that some seam cells lost their identity marker (*Figure 5E–F*). Seam cells fuse in adult *C. elegans* to form a continuous syncytium (*Sulston and Horvitz, 1977*). Using the apical junction marker AJM-1::GFP, we found that the loss of seam cell identity resulted in 'gaps' in the seam cell syncytium (*Figure 5E–F*), suggesting that *ztf-11* is capable of eliminating epithelial identity and function.

To investigate the proneural mechanism of ZTF-11 further, we tested whether ZTF-11 requires LIN-32 for its proneural activity in V1-4 lineages. Ectopic PVD-like cells could not be generated by ZTF-11 overexpression in the *lin-32(u282)* loss of function background (*Figure 5B*), suggesting that the proneural activity of ZTF-11 depends on proneural bHLH function. Consistent with the requirement for LIN-32, we found that the ectopic neurons induced by misexpression of ZTF-11 turned on *lin-32* transcriptional reporter, whereas ZTF-11::GFP-positive non-neuronal cells did not (*Figure 5—figure supplement 1*), suggesting that ZTF-11 overexpression can induce expression of LIN-32 to drive neuronal fate. In contrast, we found that ZTF-11 continued to eliminate epithelial identity in the *lin-32(u282)* mutant (*Figure 5F*). These results indicate that ZTF-11 can induce LIN-32 to specify neuronal and glial cell fate in certain circumstances. While LIN-32 promotes the 'neuronal' features, ZTF-11 helps to erase epithelial identity from prospective neuronal/glial daughters of the V5 lineage (*Figure 5G*).

## ZTF-11 is required for generating postembryonic neurons from multiple neuroectoblast lineages

Our genetic analysis revealed that ZTF-11 was important for eliminating epithelial identity during transdifferentiation of PDA neuron and neurogenesis from a neuroectoblast V5 lineage. We set out to assess whether ZTF-11 is required for neurogenesis from different postembryonic neuroectoblast lineages. We first asked whether postembryonic neurons generated during L1 larval development are present in *ztf-11(tm2315)* null mutant animals. QR/L neuroectoblast lineages contribute six postembryonic neurons (SDQR/L, AVM, PVM, AQR, and PQR) during the mid L1 stage (*Sulston and Horvitz, 1977*). Among them, AVM and PVM could be unambiguously identified as UNC-86 expressing nuclei based on their solitary positions (*Finney and Ruvkun, 1990*; *Serrano-Saiz et al., 2018*). We found that UNC-86 expression in the respective positions of AVM and PVM nuclei was invariantly lost in *ztf-11(tm2315)* late L1 animals (*Figure 6B–D*), suggesting that ZTF-11 is required for both AVM and PVM postembryonic neuronal fates. In contrast, UNC-86 expression was not lost in embryonic neurons such as ALM (*Figure 6B*). Additionally, G1 and K neuroectoblast lineages give rise to RMH and DVB neurons respectively during late L1 stage (*Sulston and Horvitz, 1977*). We again found loss of the respective cell fate markers for RMH (*Figure 6B,E*) and DVB (*Figure 6B,F*), SEM-2 (*Vidal et al., 2015*) and LIM-6 (*Hobert et al., 1999*), in *ztf-11(tm2315)* late L1 animals.

We next examined the postembryonic ventral cord motor neurons (VMNs). P1-12 (Pn) cells form the ventral epidermis of the newly hatched animal. During the late L1 stage, Pn cells give rise to postembryonic VMNs of the VA, VB, AS, VD, and VC classes (*Sulston and Horvitz, 1977*). To ask whether ZTF-11 is required for generating postembryonic VMNs, we examined condition knockouts of ZTF-11 in Pn lineages by expressing Cre in the epidermis (*Kage-Nakadai et al., 2014*). The VMNs

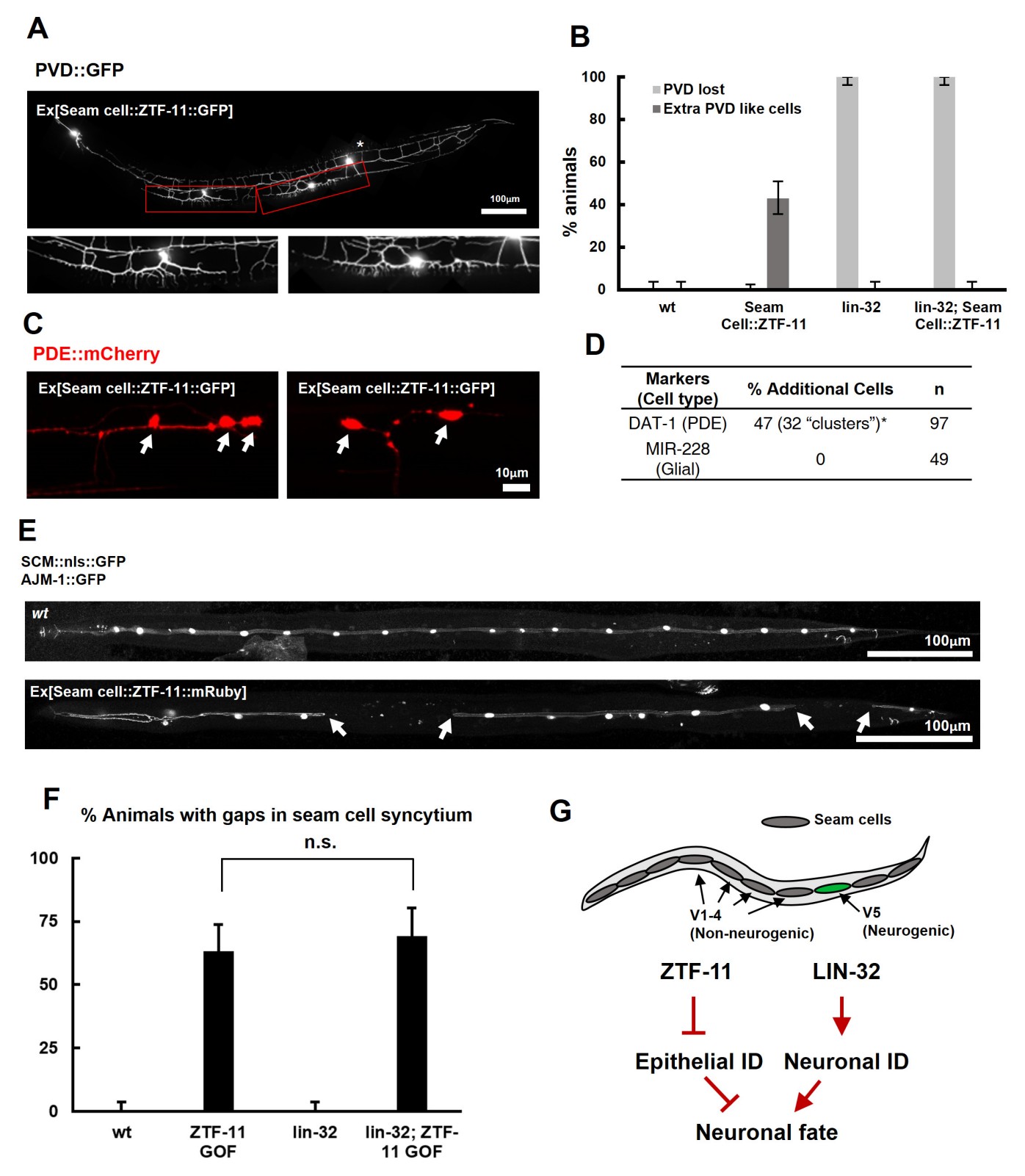

**Figure 5.** ZTF-11 is sufficient to generate neurons from epithelial cells by repressing epithelial identity. (**A**) Representative confocal image showing proneural activity of ZTF-11. ZTF-11 was expressed in seam cells resulting in epithelial seam cell lineages to produce PVD neuron-like cells. Red boxes, ZTF-11-induced PVD-like cells, zoomed in below images. Star, wild type PVD cell body. (**B**) Quantification of ZTF-11 proneural activity in wild type (wt) or *lin-32 (u282)* genetic background. *n* = 100, 158, 100, 100, respectively. Error bars are 95% Wilson-Brown C.I. See *Figure 5—source data 1* for

*Figure 5 continued on next page*

*Figure 5 continued*

numerical data. (C) Confocal images of ZTF-11 induced PDE-like cells (white arrows) from epithelial seam cell images. 'Clusters' multiple PDE-like cells were often found in close proximity (>100 µm). (D) Quantification of ZTF-11 proneural or proglial activities based on PDE marker or glial markers. (E) Representative confocal images showing seam cell identity repression by ZTF-11. ZTF-11 was expressed in seam cells resulting in repression of seam cell identities. Number of SCM-positive seam cell nuclei was reduced. Gaps in seam cell syncytium are visible with apical junction marker, *ajm-1*. (F) Quantification of seam cell syncytium gaps in wild type (wt) or *lin-32 (u282)* genetic backgrounds. Error bars are 95% Wilson-Brown C.I. *n* = 100, 111, 100, 93, respectively. n.s., p>0.05, binomial test. See *Figure 5—source data 2* for numerical data. (G) ZTF-11 and LIN-32 function in parallel to repress epithelial identity and activate neuronal identity in V5 lineage to produce postdeirid cells.

DOI: https://doi.org/10.7554/eLife.46703.018

The following source data and figure supplement are available for figure 5:

**Source data 1.** Number of counted animals of each genotype with associated phenotypes.
DOI: https://doi.org/10.7554/eLife.46703.020
**Source data 2.** Number of counted animals of each genotype with associated phenotypes.
DOI: https://doi.org/10.7554/eLife.46703.021
**Source data 3.** Raw counts of cells counted with associated phenotypes.
DOI: https://doi.org/10.7554/eLife.46703.022
**Figure supplement 1.** ZTF-11-induced ectopic neurons, but not non-neuronal cells, express LIN-32.
DOI: https://doi.org/10.7554/eLife.46703.019

can be further classified based on their respective neurotransmitters, acetylcholine, GABA, or mono-amine (serotonin). We counted the total number of VMNs expressing each neurotransmitter marker. Aminergic VMNs (two serotonergic VC4-5 neurons) are exclusively postembryonically born (*Duerr et al., 1999*; *Sulston and Horvitz, 1977*). We found that the amingergic neuron marker CAT-1 was largely lost (89% of animals) in VC4-5, suggesting that ZTF-11 is required for VC4-5 fates. In contrast to amingergic VMNs, cholinergic or GABAergic VMNs are comprised of both embryonic and postembryonic neurons (*McIntire et al., 1993*; *Pereira et al., 2015*; *Sulston and Horvitz, 1977*; *Sulston et al., 1983*). However, any loss of neuronal markers in this experiment was likely exclusively due to postembryonic neuron defects, as ZTF-11 was conditionally knocked out in only Pn lineages. With cholinergic (CHO-1) and GABAergic (UNC-47) neuronal markers, we found more subtle decreases in total cholinergic (19%) or GABAergic (5%) VMNs in *ztf-11* cKO animals. Unlike cholinergic or GABAergic postembryonic VMNs, VC4-5 neurons do not mature until the late L4 stage and maintain expression of ZTF-11 into adulthood (data not shown), which may account for their stronger requirement for ZTF-11. Taken together, our loss-of-function analysis suggest that ZTF-11 functions in multiple neuroectoblast lineages to specify postembryonic neuronal identities.

## ZTF-11 is mostly dispensable for embryonic neurogenesis but not for neuronal function

In many postembryonic lineages, neurons are generated from precursor cells, which are differentiated cells such as the rectal epithelial Y cell or the V5 precursor cell (seam cell). In contrast, the majority of embryonic neurons are generated from short-lived precursor cells through rapid cell divisions (*Sulston et al., 1983*). We next investigated the role of *ztf-11* in embryonic neurogenesis. Using a pan-neuronal RAB-3 marker, we found that most embryonic neurons are born and obtain neuronal fate in *ztf-11* mutants (*Figure 7A*). The small size of the L1 animals made it difficult to determine the exact number of RAB-3 expressing nuclei, especially amongst densely packed neurons in cephalic ganglia. We instead counted the number of embryonic motor neurons in the ventral cord, which could be unambiguously identified from *rab-3* expressing nuclei along the length of the animal (*White et al., 1976*). We found that there was a small (2%) loss of *rab-3* expressing nuclei in the ventral cord, suggesting that ZTF-11 is mostly dispensable for neuronal fate acquisition during embryogenesis (*Figure 7B*). To account for potential maternal contribution of ZTF-11, we additionally knocked down ZTF-11 in *ztf-11(tm2315)/hT2* heterozygote mothers by feeding RNAi and found that their *ztf-11(tm2315)* homozygote progeny still generated a normal number of embryonic VMNs (*Figure 7B*).

To further examine the requirement of ZTF-11 for embryonic neurogenesis, we counted the number of respective head neurons of four major neurotransmitter types (acetylcholine, glutamate, GABA, and monoamines) in wild type and *ztf-11(tm2315)* early L1 animals. Unfortunately, many

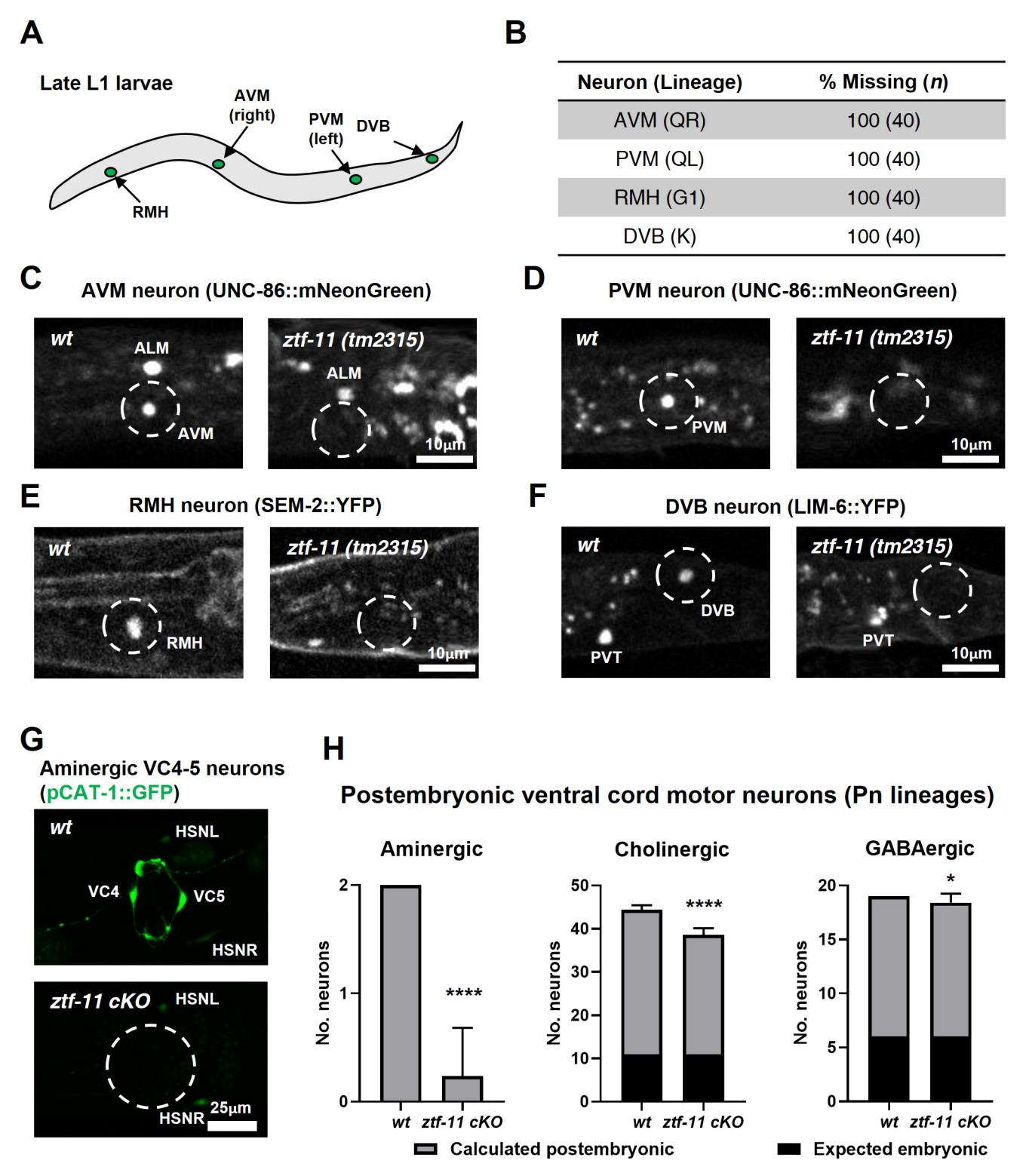

**Figure 6.** ZTF-11 is required for generating postembryonic neurons from multiple neuroectoblast lineages. (**A**) Locations of postembryonic neurons born during L1 larval development that were analyzed for this study. (**B**) Quantification of missing postembryonic neuron fate markers in *ztf-11(tm2315)* null mutants. Animals were synchronized to late L1 by bleaching and then fed for 20 hr prior to analyzing expression of respective cell fate markers (UNC-86::mNeonGreen for AVM and PVM, SEM-2::YFP for RMH, and LIM-6::YFP for DVB). (**C**) Expression of AVM marker, UNC-86, in late L1 animals of wild type or *ztf-11(tm2315)* null mutant. Dashed circle indicates the location of AVM neuron. Note that ALM, an embryonic neuron, is still generated in

*Figure 6 continued on next page*

*Figure 6 continued*

*ztf-11(tm2315)*. (**D**) Expression of PVM marker, UNC-86, in late L1 animals of wild type or *ztf-11(tm2315)* null mutant. Dashed circle indicates the location of PVM neuron. (**E**) Expression of RMH marker, SEM-2, in late L1 animals of wild type or *ztf-11(tm2315)* null mutant. Dashed circle indicates the location of RMH neuron. (**F**) Expression of DVB marker, LIM-6, in late L1 animals of wild type or *ztf-11(tm2315)* null mutant. Dashed circle indicates the location of DVB neuron. Note that PVT, an embryonic neuron, is still generated in *ztf-11(tm2315)*. (**G**) Expression of aminergic neuron marker, CAT-1, that labels VC4 and VC5 postembryonic serotonergic neurons in wild type or *ztf-11* conditional knock-out (*ztf-11 cKO*) animals. (**H**) Quantification of numbers of ventral cord motor neurons by neurotransmitter types. Neurons were counted based on markers of respective neurotransmitter types. Aminergic neurons, pCAT-1::GFP expressing cells (postembryonic: VC4-5 (two neurons)). Cholinergic neurons, pUNC-17::GFP expressing cells (postembryonic: VA2-11, VB3-11, AS2-10, and VC1-6 (34 neurons), embryonic: DA2-7 and DB3-7 (11 neurons)). GABAergic neurons, pUNC-47::GFP expressing cells (postembryonic: VD1-13 (13 neurons), embryonic DD1-6 (six neurons)). All postembryonic ventral cord motor neurons are generated from Pn lineages. As *ztf-11* was conditionally knocked-out in Pn lineages, any lost neurons were expected to be postembryonic (gray bars) rather than embryonic (black bars). Error bars are SD. ****p<0.0001, *p<0.05, Student's t-test two-tailed, *n* = 40, 119, respectively for aminergic neurons, 22, 61, respectively for cholinergic neurons, 20, 53 animals, respectively for cholinergic neurons. See *Figure 6—source data 1* for numerical data.

DOI: https://doi.org/10.7554/eLife.46703.023

The following source data is available for figure 6:

**Source data 1.** Raw counts of neurons expressing respective neurotransmitter markers in wild-type (*-cre*) or *ztf-11 cKO* animals.

DOI: https://doi.org/10.7554/eLife.46703.024

cholinergic or glutamatergic head neurons were tightly clustered in early L1 animals, which could introduce systematic errors in counting. With this caveat, we did not find significant changes in cholinergic or glutamatergic head neurons in *ztf-11(tm2315)* mutant animals, suggesting that ZTF-11 might indeed be dispensable for neuronal fates of major neurotransmitter types. Taken together, these results indicate that *ztf-11* is particularly important for neurons that are generated from epidermal lineages that have fully differentiated in both morphology and function.

Despite the near normal cell number, the *ztf-11* deletion mutants showed near complete loss of movement. When maintained with the hT2 balancer chromosome, *ztf-11(tm2315)* heterozygous mothers produced homozygous mutant individuals that were completely immobile in bacterial lawns after hatching. *ztf-11(tm2315)* mutant individuals also invariantly did not develop any further after hatching, potentially due to feeding deficits. To measure defects in motility, we performed a thrashing assay in M9 buffer. We found that homozygous *ztf-11(tm2315)* mutant individuals showed near complete loss of thrashing motion and severely uncoordinated swimming motion (*Figure 7—figure supplement 1*). In comparison, heterozygous *ztf-11(tm2315)/hT2* individuals did not show a significant change in the number of thrashes compared to wild type (N2) (*Figure 7—figure supplement 1*). These results raise the possibility that *ztf-11* may be required for proper function of embryonic neurons.

## ZTF-11 does not function through LIN-22/Hes1 repression

Next we investigated how *ztf-11* specifies neuronal fate. Previous studies suggested that proneural genes induce neuronal fate while Notch signaling inhibits neurogenesis by inhibiting proneural genes (*Bertrand et al., 2002*; *Lewis, 1998*; *Heitzler et al., 1996*; *Takebayashi et al., 1997*). Myt1 family factors are induced by proneural genes and act as transcriptional repressors of Notch signaling, including the Notch effector gene Hes1 (*Dhanesh et al., 2016*; *Mall et al., 2017*; *Vasconcelos et al., 2016*). Repressing Hes1 transcription is in turn thought to de-repress proneural bHLHs such as Ascl1, mediating exit from a proliferative neural stem cell fate and subsequent neuronal differentiation (*Mall et al., 2017*; *Vasconcelos et al., 2016*). The *C. elegans* orthologs of Hes1, *lin-22*, and its target proneural gene *lin-32*/Atoh1, function in postdeirid development (*Kenyon, 1986*; *Portman and Emmons, 2000*; *Wrischnik and Kenyon, 1997*). *lin-22*/Hes1 is expressed in seam cells, including V1-4, but not in V5 (*Katsanos et al., 2017*). In *lin-32*/Atoh1 mutants, no PVD or PDE cells were generated, while in *lin-22* mutants, additional PVD and PDE neurons were generated in each of the V1-4 lineages, suggesting that *lin-22* represses proneural gene *lin-32* in V1-4, but not in the V5 lineage (*Portman and Emmons, 2000*; *Wrischnik and Kenyon, 1997*). We set out to use this evolutionarily conserved genetic circuit to investigate whether ZTF-11 also acts through repressing *lin-22*.

We first investigated the effect of *ztf-11* cKO on *lin-22* expression by examining a transcriptional reporter for *lin-22*. Consistent with previous studies, we found that the LIN-22 transcriptional

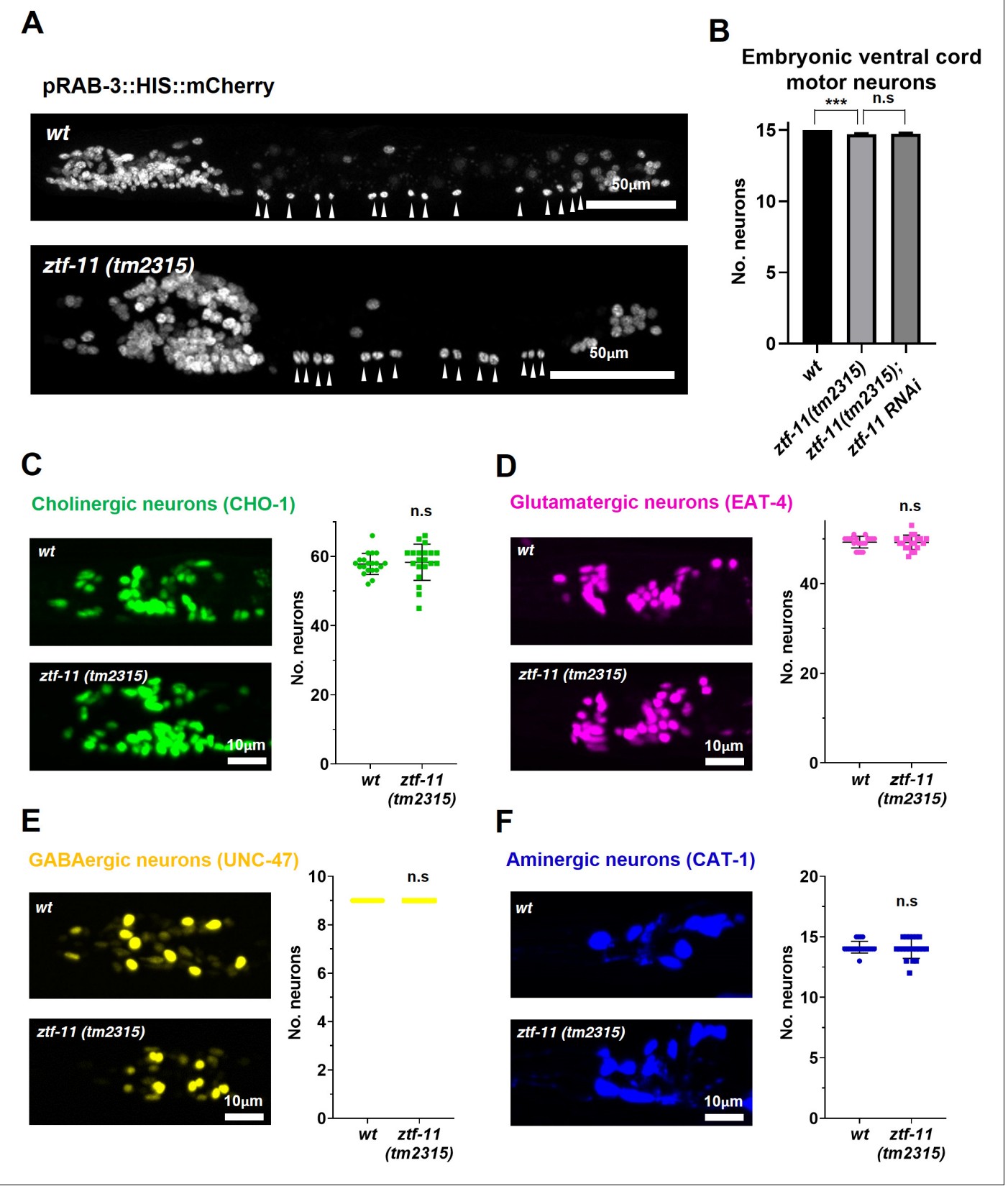

**Figure 7.** ZTF-11 is mostly dispensable for embryonic neurogenesis. (A) Pan-neuronal marker expression in wild type and *ztf-11(tm2315)* L1 larva. Arrowheads point to embryonic ventral cord motor neurons. (B) Number of embryonic ventral cord motor neurons is slightly reduced in *ztf-11(tm2315)*

*Figure 7 continued*

or *ztf-11(tm2315)* further treated with feeding RNAi against maternal ZTF-11. Neurons were counted based on pan-neuronal RAB-3 marker expression. Error bars are SD. ****p<0.0001, n.s, p>0.05 Student's t-test two-tailed, *n* = 79, 80, 50 animals respectively. See *Figure 7—source data 1* for numerical data. (C) ZTF-11 is mostly dispensable for embryonic cholinergic neurons in the head. Left, expression of cholinergic neuron marker, CHO-1, in wild type or *ztf-11(tm2315)*. Right, quantification of counted CHO-1-expressing neurons. (D) ZTF-11 is mostly dispensable for embryonic glutamatergic neurons in the head. Left, expression of glutamatergic neuron marker, EAT-4, in wild type or *ztf-11(tm2315)*. Right, quantification of counted EAT-4-expressing neurons. (E) ZTF-11 is mostly dispensable for embryonic GABAergic neurons in the head. Left, expression of GABAergic neuron marker, UNC-47, in wild type or *ztf-11(tm2315)*. Right, quantification of counted UNC47-expressing neurons. (F) ZTF-11 is mostly dispensable for embryonic aminergic neurons in the head. Left, expression of aminergic neuron marker, CAT-1, in wild type or *ztf-11(tm2315)*. Right, quantification of counted CAT-1-expressing neurons. RIH neuron was very weakly labeled by CAT-1 and only occasionally counted. (C–F) Synchronized early L1 animals by bleaching were used for experiments. Error bars are SD. n.s, p>0.05 Student's t-test two-tailed, *n* = 20 animals each. See *Figure 7—source data 2* for numerical data.

DOI: https://doi.org/10.7554/eLife.46703.025

The following source data and figure supplement are available for figure 7:

**Source data 1.** Raw counts of RAB-3 expressing nuclei in animals of each genotypes.
DOI: https://doi.org/10.7554/eLife.46703.027
**Source data 2.** Raw counts of nuclei expressing respective neurotransmitter markers in wild-type or *ztf-11(tm2315)* animals.
DOI: https://doi.org/10.7554/eLife.46703.028
**Source data 3.** Raw counts of thrashes exhibited by animals of each genotypes.
DOI: https://doi.org/10.7554/eLife.46703.029
**Figure supplement 1.** Embryonic ZTF-11 is required for coordinated motility.
DOI: https://doi.org/10.7554/eLife.46703.026

reporter was invariantly excluded from V5 lineage cells during postdeirid development (*Figure 8B*, top panel; *Figure 8C*). However, in the *ztf-11* cKO, the LIN-22 reporter remained undetectable in V5 lineages. In addition, we detected no change in the expression pattern or level of the LIN-22 reporter in V1-4 (*Figure 8B*, bottom panel; *Figure 8C*). This result suggests that *lin-22* is unlikely to be a transcriptional target of ZTF-11.

To additionally test whether ZTF-11 functions through repression of LIN-22 activity, we performed a genetic analysis of *ztf-11* cKO and *lin-22(n372)*. The *lin-22(n372)* single mutant generates ectopic neurons in the V1-4 lineages. If *ztf-11* acts upstream to repress *lin-22*, we predicted that *ztf-11* cKO; *lin-22* double mutant would show a *lin-22* single mutant phenotype, with ectopic neurons generated from the V1-4 lineages. Contrary to this prediction, we observed that *ztf-11* cKO; *lin-22* double mutant resulted in a loss of the PVD cell marker in V1-4 as well as V5 lineages (*Figure 8D–E*). Both the genetic and expression data argue against the proposed model in which ZTF-11 acts via repression of LIN-22.

Next, we investigated whether ZTF-11 de-represses *lin-32* transcription. To understand *ztf-11*'s relationship with *lin-32*, we examined *lin-32* expression in *ztf-11* cKO mutant and found that *lin-32* is still expressed in the V5 lineage at the level of wild-type controls (*Figure 8F–G*). However, ZTF-11 expression in the V5 lineage is completely eliminated in the *lin-32* mutant (*Figure 8A*), suggesting that *ztf-11* is turned on by LIN-32. This is consistent with the fact that the E-box sequences in the promoter region of *ztf-11* are required for its expression (*Figure 2*). Additionally, the loss of *lin-32* did not alter the ability of ectopically expressed ZTF-11 to reprogram epithelial identity (*Figure 5B*), suggesting that *ztf-11* acts downstream of both *lin-22* and *lin-32* to repress epithelial fate. We next measured the fluorescence intensity of *lin-32* transcriptional reporter during postdeirid development. If ZTF-11 acts as a de-repressor of *lin-32*, we would expect a loss or reduction of *lin-32* transcriptional activity. However, we did not observe a significant change in fluorescence intensity between wild type and *ztf-11* cKO. Together, these results argue against the existing model in which *ztf-11* acts through repressing *lin-22* and instead support a linear genetic model in which *lin-22*/Hes1 represses *lin-32*/Atoh1, which in turn activates *ztf-11* (*Figure 8A*).

## ZTF-11 negatively regulates non-neuronal genes

To understand how *ztf-11* promotes neuronal fate, we performed transcriptome analysis in *ztf-11* knockdown animals during development. RNAi hypersensitive *eri-1(mg366); ztf-11::gfp* worms were fed with bacteria expressing dsRNA against ZTF-11 (Ahringer RNAi collection) or empty feeding

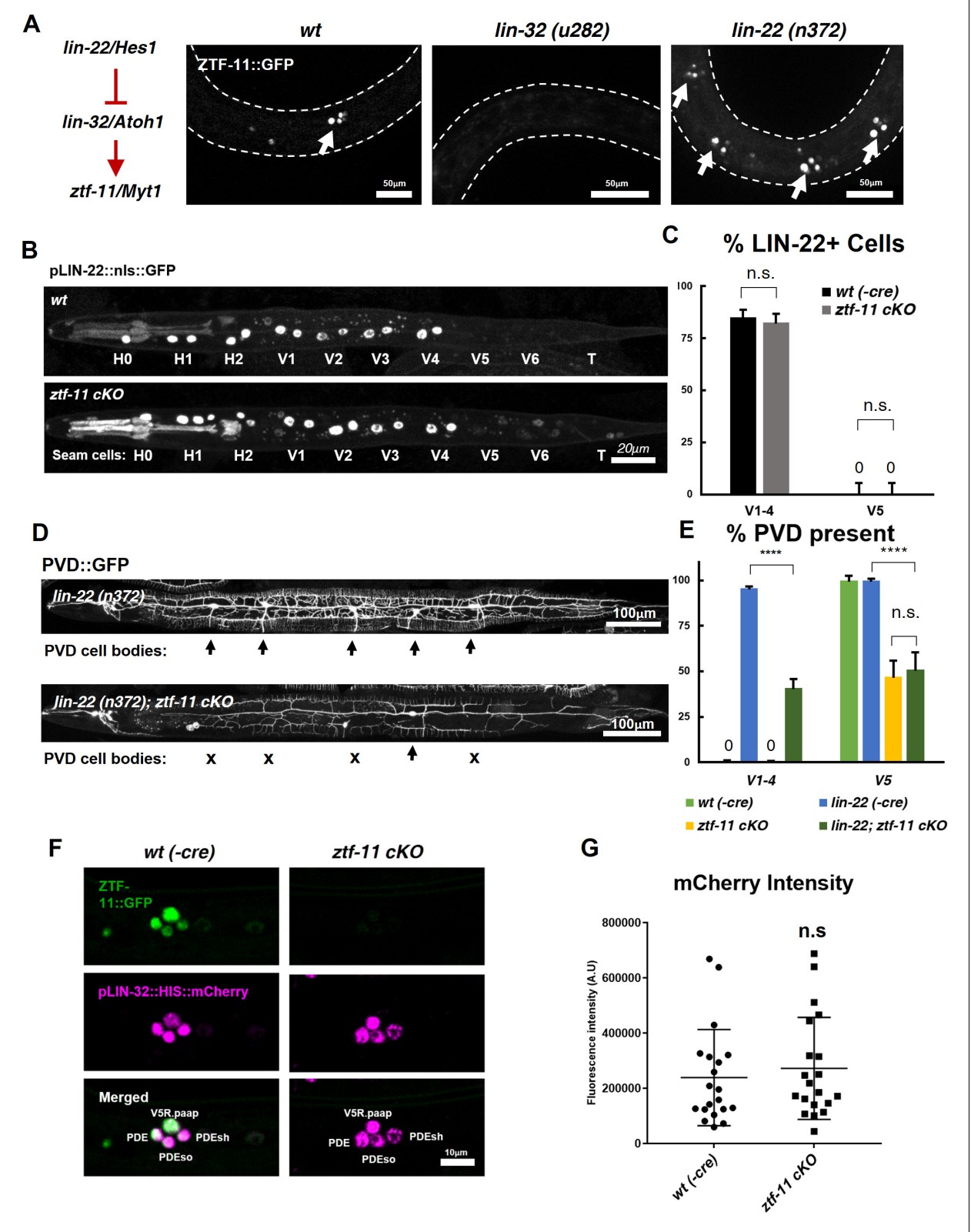

**Figure 8.** ZTF-11 does not function by repressing *lin-22*/Hes1. (**A**) *ztf-11* functions downstream of postdeirid development genes *lin-22* and *lin-32*. Right, representative images of ZTF-11::GFP expression during postdeirid neurogenesis in wild type, *lin-32*, or *lin-22* mutants. (**B**) *lin-22* transcriptional reporter expression in wild type and *ztf-11* cKO early L2 larvae. LIN-22 is expressed in lateral seam cells but excluded from three posterior seam cells (V5, V6, and T) Pharyngeal staining is bleed through from co-injection marker pMYO-2::mCherry. (**C**) Quantification of pLIN-22::GFP-positive seam cells

*Figure 8 continued on next page*

*Figure 8 continued*

in b. *n* = 66–68 animals each. Error bars, 95% Wilson-Brown C.I. Fischer's exact test n.s, p>0.05. See *Figure 7—source data 1* for numerical data. (D) *ztf-11* cKO is epistatic to *lin-22(n372)*, suggesting that ZTF-11 is unlikely to function through repressing LIN-22. *lin-22(n372)* results in duplication of postdeirid lineage in V1-4, resulting in five postdeirid on each side. In *ztf-11* cKO *lin-22(n372)* double mutant, some of four additional postdeirid as well as wild type V5 postdeirid, are lost. (E) Quantification of PVD::GFP-positive cells from d. Parent seam cell lineage for each PVD::GFP positive cells were inferred from position along AP axis. *n* = 100–101 animals each. Error bars, 95% Wilson-Brown C.I. Fischer's exact test, ****p<0.0001. n.s., p>0.05. See *Figure 7—source data 2* for numerical data. (F) *lin-32* transcriptional marker expression in wild type and *ztf-11* cKO. (G) Quantification of *lin-32* transcriptional activity. pLIN-32::HIS::mCherry fluorescence intensity was scored. *n* = 20 animals each. Error bars, S.D. Student's t-test two-tailed, n.s, p>0.05. See *Figure 7—source data 3* for numerical data.

DOI: https://doi.org/10.7554/eLife.46703.030

The following source data is available for figure 8:

**Source data 1.** Raw counts of LIN-22 expression in V cells in wild-type of *ztf-11 cKO* animals.
DOI: https://doi.org/10.7554/eLife.46703.031
**Source data 2.** Raw counts of PVD or PVD-like cells in animals of each genotype.
DOI: https://doi.org/10.7554/eLife.46703.032
**Source data 3.** Raw fluorescence intensity values of LIN-32 transcriptional reporter in wild-type(-Cre) or *ztf-11 cKO* animals.
DOI: https://doi.org/10.7554/eLife.46703.033

RNAi vector as a control. *ztf-11::gfp* fluorescence was strongly reduced in embryos fed with *ztf-11* RNAi, confirming the knockdown efficiency (data not shown). Consistent with this observed reduction of ZTF-11::GFP, we found that *ztf-11* transcript levels were reduced by 72% in *ztf-11* knockdown embryos (*Supplementary file 2*).

Differential expression analysis revealed that 419 genes were significantly dysregulated in *ztf-11* KD embryos (FDR < 0.1) (*Figure 9A*; *Supplementary file 2*). The majority (88%) of the differentially expressed genes were upregulated in *ztf-11* KD, consistent with the hypothesis that *ztf-11* acts a transcriptional repressor (*Figure 6A*). Notably, among the upregulated genes, the vast majority were non-neuronal genes, including genes specific to epidermis (collagens) or muscle (sarcomere components) (*Figure 9B*). GO-term enrichment analysis revealed that epidermal and muscular genes were significantly enriched among upregulated genes. In contrast, we did not find significant changes in expression of most neuronal genes, including those involved in neurodevelopment, synaptic transmission, axon guidance, and neurotransmitter synthesis (*Figure 9C*), likely reflecting our findings that embryonic neurons are still generated in *ztf-11* null mutant. These findings are consistent with our genetic analysis, which demonstrated that ZTF-11 acts as a repressor of epithelial identity rather than a direct driver of neuronal fate.

## ZTF-11 mediates transcriptional repression through binding with MuvB co-repressor complex

Our transcriptomic analysis suggested that ZTF-11 mostly represses gene expression. To further test if ZTF-11 indeed functions as a transcriptional repressor, we fused its DNA binding Zinc-finger (ZF) domains with either a transcriptional activator (VP64) or repressor (EnR) domain. We found that the expression of the transcriptional repressor fusion protein (EnR::ZF) in seam cells resulted in proneural activity similar to overexpression of the native ZTF-11 protein. In contrast, expression of the transcriptional activator fusion protein (VP64::ZF) showed a dominant negative effect and blocked postdeirid neurogenesis (*Figure 10A*). Based on these results, we conclude that ZTF-11, like vertebrate Myt1 family proteins, indeed functions as a transcriptional repressor to promote neuronal fate (*Mall et al., 2017*; *Vasconcelos et al., 2016*).

Transcription factors repress gene expression by recruiting corepressor complexes. Corepressor complexes modify chromatin into a more repressed state by catalyzing posttranslational modification of histone tails. Histone chaperones RbAp46/48 mediate interaction with the histone and thus form the core histone-binding subunits of several histone post-translational modifying complexes (*Huang et al., 1991*; *Loyola and Almouzni, 2004*; *Qian et al., 1993*). To investigate the mechanism of transcriptional repression by ZTF-11, we first examined the role of histone chaperone RbAp46/48 homologs, RBA-1 and LIN-53, in postdeirid neurogenesis where ZTF-11 is required. In *rba-1* or *lin-53* single mutants, approximately 20% of the PVD neurons are missing while another 40% of PVDs showed severe morphological defects (*Figure 10C–D*). This result suggests that the histone

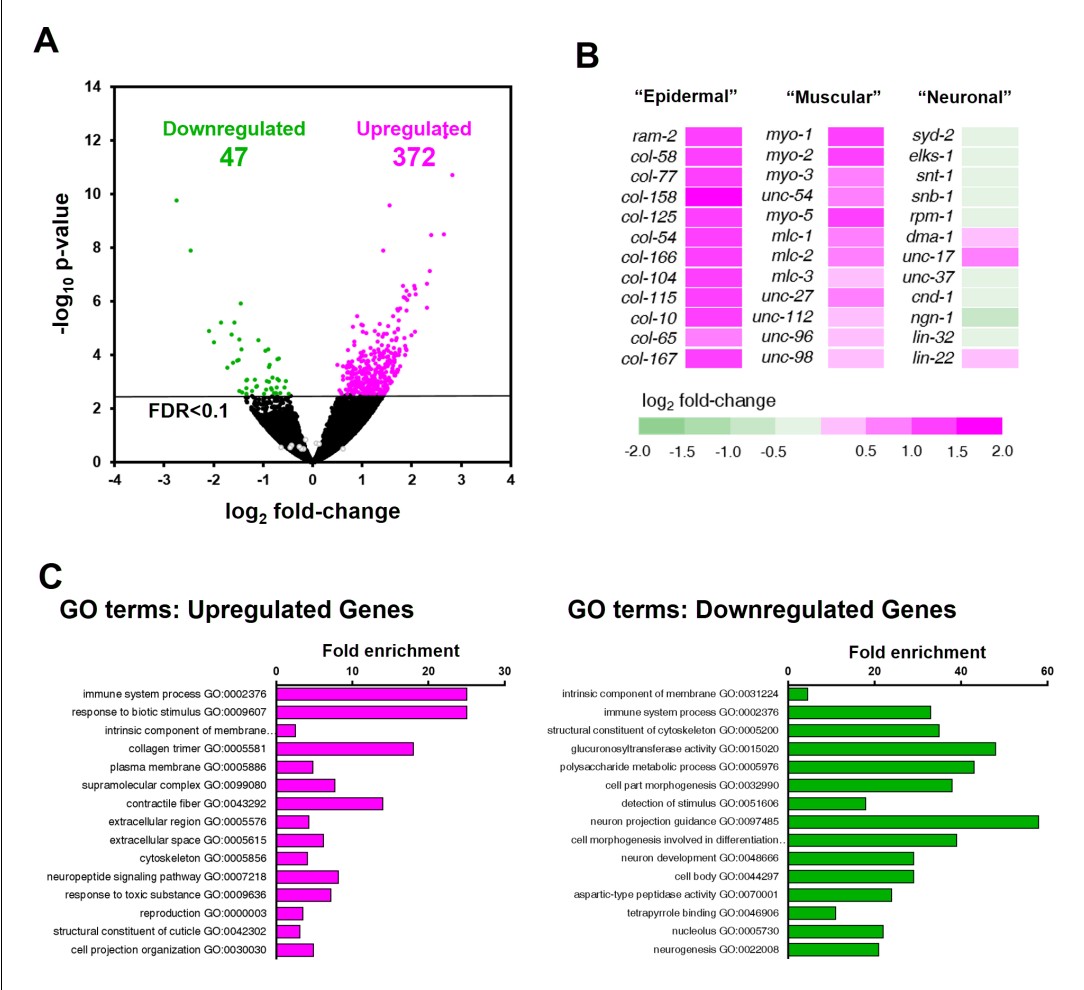

**Figure 9.** ZTF-11 negatively regulates non-neuronal genes. (A) Volcano plot of differentially expressed genes in *ztf-11*-depleted embryos. Whole embryos treated with RNAi against *ztf-11* or control vector were used for the experiments. Numbers, number of genes that were significantly (FDR < 0.1) upregulated (magenta) or downregulated (green). Selected neuronal genes from (B) are shown as gray circles. See *Supplementary file 2* for full list of differentially expressed genes. (B) Heat map showing expression level changes in selected cell-type identity markers. Epidermal identity genes were curated from among the significantly changed collagen genes. Muscle (sarcomere) and neuronal identity genes were chosen blind to the fold changes. (C) GO-term enrichment analysis using gene sets of significantly upregulated (magenta) or downregulated (green) transcripts. Note that epidermal (collagen trimer, structural component of cuticle) and muscular (contractile fiber) terms are enriched in upregulated gene set. See *Supplementary file 2* for full list of enriched GO-terms.

DOI: https://doi.org/10.7554/eLife.46703.034

chaperone RbAp46/68 homologs, *rba-1* and *lin-53*, were required for proper postdeirid neurogenesis and that ZTF-11 likely functions through a corepressor complex containing histone chaperones RbAp46/48.

Vertebrate Myt1 interacts with the Sin3 histone deacetylase corepressor complex (Sin3-HDAC), which contains RbAp46/48, to repress target genes during transdifferentiation in vitro (*Mall et al., 2017*; *Romm et al., 2005*). It is unclear whether Myt1 family factors also function with the Sin3-HDAC complex in developmental contexts. We examined the role of Sin3-HDAC components in postdeirid neurogenesis and observed no defects resulting from the loss of *sin-3*, the sole Sin3 homolog in worms (*Choy et al., 2007*) (*Figure 10C–D*). We next tested whether other corepressor complexes that contain RbAp46/48 are involved in postdeirid neurogenesis. We found components of the MuvB core of the DRM(*DP/Rb/M*uvB) corepressor complex, *lin-9*, *lin-52*, and *lin-54* (*Harrison et al., 2006*), to be required for robust postdeirid neurogenesis (*Figure 10C–E* and *Figure 10—figure supplement 1*). These genetic results suggest that the MuvB repressor complex,

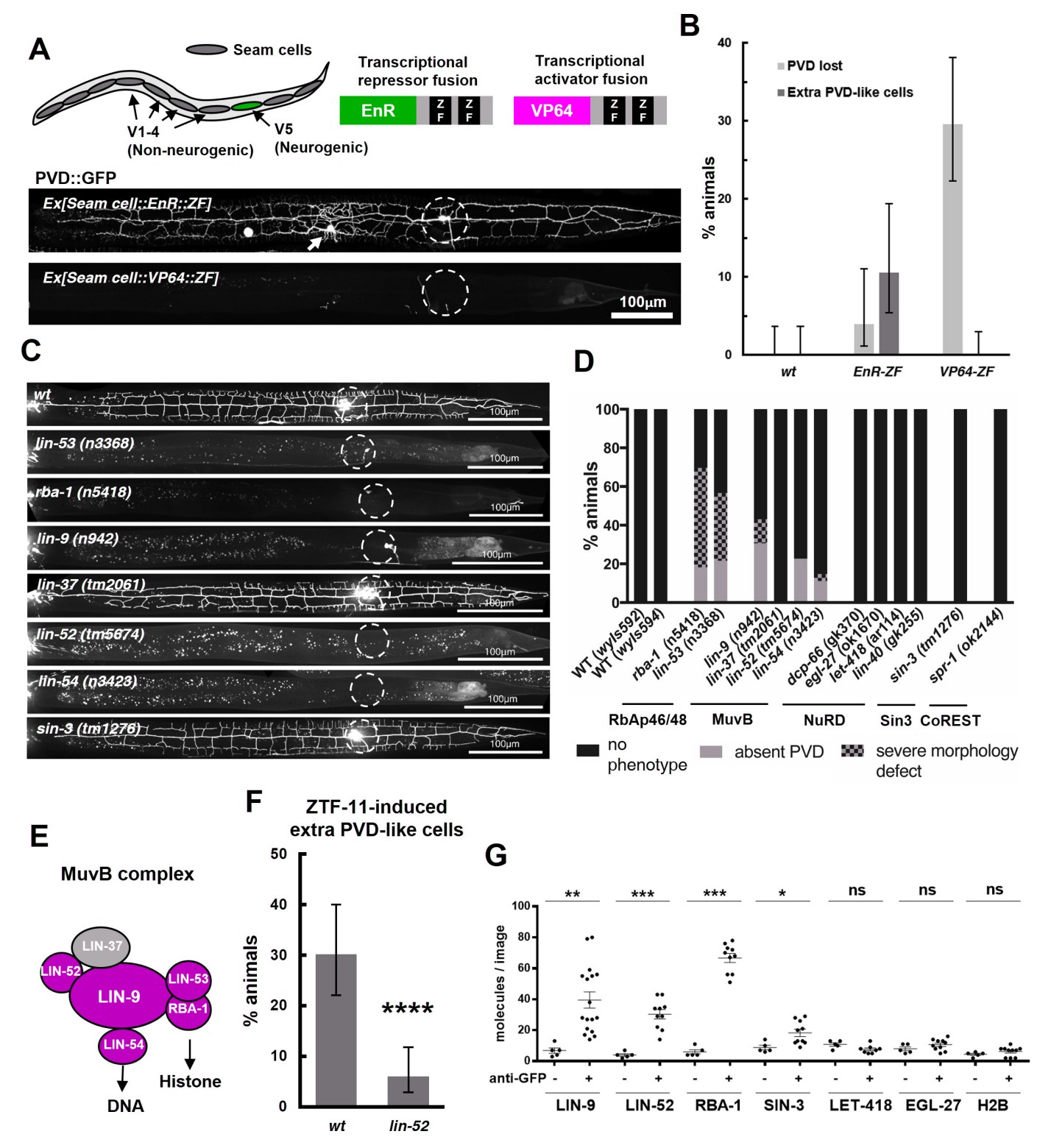

**Figure 10.** ZTF-11 functions with the MuvB co-repressor complex. (**A**) Top, schematic diagram of the experiment. Transcriptional repressor (EnR-ZF) or activator (VP64-ZF) fusion proteins were expressed in both neurogenic V5 and non-neurogenic V1-4 seam cells. Bottom, representative confocal images showing extra PVD-like cells generated with EnR-ZF expression or loss of PVD generated with VP64-ZF expression. (**B**) Quantification of respective fusion protein overexpression phenotypes. Error bars are 95% Wilson-Brown C.I. *n* = 100, 76, 125 for wt, EnR-ZF, VP64-ZF, respectively. See *Figure 10— source data 1* for numerical data. (**C**) Representative confocal images showing postdeirid neurogenesis phenotypes in wild type, MuvB, or Sin3 co-

*Figure 10 continued on next page*

*Figure 10 continued*

repressor complex mutants. (D) Quantifications of PVD neurogenesis defects of MuvB, Sin3, NuRD, or CoRest co-repressor complexes. *n* = 27–117. (E) Schematic diagram of core components of MuvB complex. Genes required for postdeirid neurogenesis are colored in magenta. See also *Figure 10— figure supplement 1* for fly and human orthologs. (F) Proneural activity of *ztf-11* requires MuvB gene *lin-52*. ZTF-11 was ectopically expressed in seam cells in wild type (N2) or *lin-52(tm5674)* backgrounds and transgenic animals were scored for presence of ectopic PVD-like cells. Error bars are 95% Wilson-Brown C.I. *n* = 96, 117, respectively. Binomial test, ****p<0.0001. See *Figure 10—source data 2* for numerical data. (G) SiMPull experiment shows binding of MuvB complex components to ZTF-11. Error bars, S.E.M. *n* = 5–17 each. Student's t-test two-tailed *p<0.05, **p<0.005, ***p<0.0005. See *Figure 10—source data 3* for numerical data.

DOI: https://doi.org/10.7554/eLife.46703.035

The following source data and figure supplement are available for figure 10:

**Source data 1.** Number of animals with associated phenotypes in wild-type or transgenic lines expressing transcriptional repressor (EnR-ZF) or activator (VP64-ZF) fusion proteins.
DOI: https://doi.org/10.7554/eLife.46703.037
**Source data 2.** Number of counted animals of each genotype with associated phenotypes.
DOI: https://doi.org/10.7554/eLife.46703.038
**Source data 3.** Number of mCherry-tagged corepressor molecules in each raw images.
DOI: https://doi.org/10.7554/eLife.46703.039
**Figure supplement 1.** MuvB complex genes and their homologs in model organisms.
DOI: https://doi.org/10.7554/eLife.46703.036

rather than the Sin3-HDAC complex, is required for V5 neurogenesis in vivo. Consistent with this result, the proneural activity of ectopically expressed ZTF-11 is strongly (80%) reduced by the loss of *lin-52*, suggesting that ZTF-11 functions through the MuvB complex (*Figure 10F*).

We next tested whether ZTF-11 directly binds to MuvB complex components using a single molecule pull-down (SiMPull) analysis, a quantitative, imaging-based co-immunoprecipitation assay (*Jain et al., 2011*). We used CRISPR/Cas9 to endogenously tag candidate co-repressor subunits with mCherry, and *ztf-11* was tagged with GFP. ZTF-11::GFP was pulled down by anti-GFP antibody and displayed on a microscopy slide. Co-precipitation was quantified by counting the number of mCherry-tagged co-repressor molecules on the slide. We found that LIN-9, LIN-52, and the RbAp46/48 homolog RBA-1 could be pulled down by ZTF-11, suggesting that ZTF-11 binds to DRM co-repressor complex in vivo (*Figure 10G*). Interestingly, *lin-37* mutant did not show obvious neurogenesis defects (*Figure 10C–D*), suggesting that LIN-37 may be dispensable for association of MuvB complex with ZTF-11. Unexpectedly, we found weak co-precipitation of SIN-3 with ZTF-11::GFP, suggesting that ZTF-11 may still function with SIN-3 in other in vivo contexts. Taken together, these results also provide the first evidence that the evolutionarily conserved MuvB co-repressor complex (LINC in vertebrates and dREAM/MuvB complex in flies, see *Figure 10—figure supplement 1*) (*Lewis et al., 2004*; *Schmit et al., 2007*) functions in neurogenesis. It has previously been shown that MuvB genes are required to repress germline genes in somatic cells (*Petrella et al., 2011*; *Wang et al., 2005*). We propose that ZTF-11 binds to MuvB complex to similarly repress many non-neuronal genes during neuronal development.

## Discussion

Although Myt1 family factors were discovered more than two decades ago, the neurogenic role of Myt1 family factors is only starting to be unraveled on a molecular level. In particular, advances in in vitro transdifferentiation have provided crucial insights in recognizing Myt1 family factors as key drivers of neurogenesis. Here we studied the physiological and developmental functions of ZTF-11 in neurogenesis in vivo. Comparing our results to the in vitro transdifferentiation literature, there are interesting similarities and differences. First, in both systems, ZTF-11 and Myt1l are critical drivers of neural cell fate. Examining neurogenesis in different classes of neurons in *C. elegans*, we found that ZTF-11 is particularly important for the generation of postembryonic neurons, which are derived from epidermal cells. Most interestingly, a developmentally occurring epithelial-to-neuronal transdifferentiation event requires ZTF-11 to reprogram epithelial identity, further bridging the in vivo and in vitro neurogenic functions of Myt1 factors. In contrast, embryonic neurons generated by rapid cell division from precursor cells are less dependent on ZTF-11. It is conceivable that ZTF-11 is required

to 'turn off' the established epidermal/epithelial cell fate before neurons can be generated. This could explain the particular requirement of Myt1l in transdifferentiation. Second, both ZTF-11 and Myt1l function as transcriptional repressors. This was evident from RNA-seq experiments from both *C. elegans* and transdifferentiating mammalian neurons (*Mall et al., 2017*), where non-neuronal genes are suppressed by this family of proteins. Third, the neuronal expression of ZTF-11 in developing neurons is activated by proneural bHLH genes, which is likely dependent on the conserved E-boxes in the promoter elements of *ztf-11* and Myt1l. Together, these similarities build a strong case that the Myt1 family transcription factors play conserved functions in neuronal specification by repressing the expression of non-neuronal genes. These results also suggest that repression of non-neuronal genes is an important aspect of neurogenesis across species.

We also identified two key differences between ZTF-11 and Myt1. First, our genetic analysis shows that ZTF-11 does not repress the Hes1 homolog *lin-22*. How can this result be reconciled with previous results that Myt1 family factors repress lateral inhibition? One possible explanation for this discrepancy is that ZTF-11 may have lost its ability to repress Notch signaling. However, it is also possible that vertebrate Myt1 family factors gained the ability to repress Hes1. Second, our data also suggest that the MuvB complex, but not the Sin3-HDAC complex, plays an important role in neurogenesis as a co-repressor complex that functions with ZTF-11. This result is interesting but unsurprising considering the sequence divergence of Myt1 family proteins outside the conserved DNA-binding zinc-fingers (*Figure 1—figure supplement 1*). Myt1 family proteins may have diverged through evolution to function with different co-repressor complexes. It is also noteworthy that ZTF-11 retained weak binding with SIN-3 (*Figure 10G*), suggesting this interaction is conserved through evolution in other developmental contexts. Examination of Myt1 family factors in other invertebrate model systems is likely to shed light on these intriguing questions.

During evolution, ancestral neurons likely arose from non-neuronal cells. Consistent with this hypothesis, cnidarian neurons are generated from endodermal interstitial stem cells or epithelial precursors, rather than dedicated neural precursors (*Rentzsch et al., 2017*). As with proneural genes, Myt1 family factors are conserved throughout metazoan evolution with the exception of porifera (sponges) and ctenophora (comb jellies), which either lack a nervous system or are thought to have independently evolved a nervous system (*Moroz et al., 2014*). MuvB complex genes are conserved in all animals regardless of presence of the nervous system, suggesting that Myt1 family proteins evolved later and recruited MuvB as their co-repressor. It is now tempting to speculate that Myt1 family factors, alongside MuvB co-repressor complex, may comprise an ancestral core module for generating neurons from non-neuronal cells.

## Materials and methods

**Key resources table**

| Reagent type (species) or resource | Designation | Source or reference | Identifiers | Additional information |
|---|---|---|---|---|
| Gene (*Caenorhabditis elegans*) | *ztf-11* | NA | Wormbase gene: WBGene00009939 | |
| Strain, strain background (*C. elegans*) | N2 | C. elegans Genetic Center (CGC) | | Wild type strain |
| Genetic reagent (*C. elegans*) | *ztf-11(tm2315)* | National Bioresource Project (Dr. Shohei Mitani) | | |
| Genetic reagent (*C. elegans*) | *lin-32(u282)* | C. elegans Genetic Center (CGC) | | |
| Genetic reagent (*C. elegans*) | *lin-22(n372)* | C. elegans Genetic Center (CGC) | | |

*Continued on next page*

*Continued*

| Reagent type (species) or resource | Designation | Source or reference | Identifiers | Additional information |
|---|---|---|---|---|
| Genetic reagent (*C. elegans*) | *lin-52(tm5674)* | National Bioresource Project (Dr. Shohei Mitani) | | |
| Genetic reagent (*C. elegans*) | *ztf-11(wy1077)* | This study | | ZTF-11::GFP endogenous knock-in |
| Genetic reagent (*C. elegans*) | *ztf-11(wy1088)* | This study | | Floxed *ztf-11* allele |
| Genetic reagent (*C. elegans*) | *ztf-11(wy1100)* | This study | | Floxed *ztf-11::gfp* allele |
| Genetic reagent (*C. elegans*) | *lin-9(wy1224)* | This study | | mCherry::LIN-9 endogenous knock-in |
| Genetic reagent (*C. elegans*) | *lin-52(wy1225)* | This study | | LIN-52::mCherry endogenous knock-in |
| Genetic reagent (*C. elegans*) | *rba-1(wy1212)* | This study | | RBA-1::mCherry endogenous knock-in |
| Genetic reagent (*C. elegans*) | *sin-3(wy1210)* | This study | | SIN-3::mCherry endogenous knock-in |
| Genetic reagent (*C. elegans*) | *let-418(wy1215)* | This study | | LET-418::mCherry endogenous knock-in |
| Genetic reagent (*C. elegans*) | *egl-27(wy1207)* | This study | | EGL-27::mCherry endogenous knock-in |
| Transfected construct (*E. coli* HT115 (DE3)) | Feeding RNAi clone against *ztf-11* | Dr. Julie Ahringer, Source BioScience | RRID:SCR_017064 | Primer pair number: 1528 |
| Antibody | anti-GFP, biotin conjugated (Rabbit polyclonal) | Rockland Immunochemicals | Rockland Cat# 600-406-215, RRID:AB_828168 | |
| Recombinant DNA reagent | *pnhr-81::ztf-11::GFP* | This paper | | ZTF-11 seam cell gain of function |
| Recombinant DNA reagent | *pnhr-81::ztf-11::mRuby3* | This paper | | ZTF-11 seam cell gain of function (used in conjunction with seam cell markers) |
| Recombinant DNA reagent | *pztf-11::his::mCherry* | This paper | | *ztf-11* transcriptional reporter |
| Recombinant DNA reagent | *pztf-11::his::GFP* | This paper | | *ztf-11* transcriptional reporter |
| Recombinant DNA reagent | *pztf-11::his::GFP(-Ebox)* | This paper | | E-box mutated *ztf-11* transcriptional reporter |
| Recombinant DNA reagent | *pegl-26::Cre* | This paper | | Rectal epithelial Cre |
| Recombinant DNA reagent | *pnhr-81::Cre* | This paper | | Seam cell Cre |
| Recombinant DNA reagent | *pnhr-81::vp64::ztf-11(217-360)* | This paper | | Transcriptional activator fusion |

*Continued on next page*

*Continued*

| Reagent type (species) or resource | Designation | Source or reference | Identifiers | Additional information |
|---|---|---|---|---|
| Recombinant DNA reagent | *pnhr-81::EnR:: ztf-11(217-360)* | This paper | | Transcriptional repressor fusion |
| Commercial assay or kit | RNeasy Plus Micro Kit | Qiagen | Cat#: 74034 | |
| Commercial assay or kit | QiaShredder | Qiagen | Cat#: 79654 | |
| Chemical compound, drug | Chymotrypsin | Sigma Aldrich | Cat#: CHY5S | |
| Chemical compound, drug | Chitinase | Sigma Aldrich | Cat#: C6137 | |
| Software, algorithm | ImageJ | NIH | RRID:SCR_003070 | |
| Software, algorithm | GraphPad Prism | GraphPad | RRID:SCR_002798 | |

All sequencing dataset generated during this study is available on NCBI GEO (Accession code: GSE125694). All materials generated and analyzed during the current study are available from the corresponding author on reasonable request.

## Nematode culture and strains

Wild-type strains were *C. elegans* variety Bristol, strain N2. Worms were maintained by standard methods as previously described (*Brenner, 1974*). Worms were grown at 20°C on nematode growth media (NGM) plates seeded with bacteria (*Escherichia coli* OP50) as a food source. Transgenic strains were generated as previously described by gonadal injection (*Mello and Fire, 1995*). The epidermal Cre strain (FX15987) was kindly provided by Dr. Shohei Mitani. The list of all mutant and transgenic strains used in this study is available in *Supplementary file 3*.

## Cloning and constructs

DNA plasmid constructs were generated by PCR amplification using Pfusion DNA polymerase followed by isothermal assembly or restriction digest and subsequent ligation using T4 DNA ligase (NEB). *ztf-11* cDNA was amplified using *C. elegans* ORFeome library (*Lamesch et al., 2004*). *ztf-11* promoter (pZTF-11) was cloned via PCR amplification of 2740 bp fragment upstream of *ztf-11 tss*. pZTF-11 ΔE-box mutations were generated with gBlock synthesis (IDT) followed by isothermal assembly into wild type pZTF-11 vector. EnR and VP64 DNA were kindly provided by Dr. Mauritz Mall. Unless otherwise indicated, worm lysate genomic DNA was used as the template for PCR amplification. A complete list of DNA constructs and oligos is available in *Supplementary file 4*.

## CRISPR/Cas9 genome editing

Two CRISPR/Cas9 genome editing protocols were used for this study. In brief, for insertion of GFP or mCherry into endogenous genetic loci, GFP or mCherry was PCR amplified with primers containing homology arms for insertion sites as donor DNA for homologous recombination. Donor DNA was co-injected with gRNA (IDT), crRNA (IDT), and Cas9 enzyme (IDT) as previously described (*Paix et al., 2017*). F1 generation animals were visually screened for presence of GFP or mCherry signals using Axioplan2 Fluorescence microscope (Carl Zeiss). GFP or mCherry-positive animals were then homozygosed in F2 generation and verified by Sanger sequencing. For insertion of loxP sites into *ztf-11* locus, a co-conversion strategy was used as previously described (*Arribere et al., 2014*). Synthesized loxP sequence ssDNA with 60 bp homology arms flanking insertion sites (IDT) was used as donor DNA for homologous recombination. Donor DNA was co-injected with Cas9 expressing plasmid (pJW1259, kindly provided by Dr. Jordan Ward), sgRNA expressing vectors, dpy-10 targeting-gRNA, and dpy-10 donor DNA. F1 animals were screened by PCR amplification of the loxP inserts.

## 4-D imaging and lineage tracing

Embryos were collected from gravid hermaphrodites and mounted with polystyrene beads (Polysciences Inc) as described (*Du et al., 2015*). Embryos were imaged on a Zeiss AxioObserver Z1 inverted microscope frame with Yokogawa CSU-X1 spinning disk and an Olympus UPLSAPO 60xs silicone oil immersion objective. GFP and mCherry channels were acquired simultaneously on a pair of aligned EMCCD cameras (C9100-13). Image acquisition was performed using MetaMorph software (Molecular Devices). Embryos were imaged every 75 s, with 30 z slices at 1 μm apart. Lineage tracing and quantification of marker expression were done with the StarryNite and AceTree software as described (*Du et al., 2015*).

## Microscopy

Hermaphrodite animals were anesthetized using 2.5 mM levamisole, mounted on 3% agar pads, and imaged using a Zeiss LSM710 confocal microscope (Carl Zeiss) with a Plan-Apochromat 40x/1.3 NA objective or 63x/1.4NA objective. Z stacks and maximum-intensity projections were generated using ImageJ (NIH). The imaging was not done by an experimenter blind to the experimental condition. Fluorescence intensity measurements (*Figure 2*, *Figure 8*, and *Figure 4—figure supplement 2*) were quantified using ImageJ (NIH). Quantification for cell identity markers were performed using Axioplan2 fluorescence microscope (Carl Zeiss) with a Plan-Apochromat 40x/1.3 NA objective. When quantifying any cKO animals, presence of Cre-expressing transgene was checked after each animal's phenotype was determined to prevent potential bias.

## L1 thrashing experiment

Worms grown on NGM plates were transferred to room temperature at least 1 hr prior to the assay. Individual L1 animals were carefully transferred to a small drop of M9 media on a glass slide. Following a minute of incubation in M9, the number of thrashing events (defined by one cycle of alternating 'C' bends) were then counted for a minute. The genotype of each assayed animals were determined after each counting to circumvent potential bias.

## RNA-seq sample preparation

Samples for RNA-seq was prepared by feeding RNAi (*Timmons et al., 2001*). In brief, animals were harvested from NGM plates and eggs were collected by bleaching. Eggs were hatched overnight in M9 media to get synchronized L1 larval culture (*Porta-de-la-Riva et al., 2012*). Synchronized L1 cultures were inoculated on plates expressing feeding RNAi clones for *ztf-11* (*Kamath and Ahringer, 2003*) (Dr. Julie Ahringer) or a control vector (L4440). After 64 hr, adult worms bearing eggs were harvested and eggs were collected by careful bleaching. Eggs were incubated in M9 for 3 hr to allow them to develop into gastrula stages. Egg shell was disturbed with chymotrypsin (Sigma-Aldrich) and chitinase (Sigma-Aldrich) as previously described (*Edgar and Goldstein, 2012*) and lysed by centrifuging through QiaShredder columns (Qiagen) following the manufacturer's instructions. RNA was isolated from the eggs using RNeasy Plus Micro Kit (Qiagen) following the manufacturer's instructions. Resulting RNA samples were quality controlled by Agilent Bioanalyzer 2100 and only the RNA samples with RIN of 9 or higher were submitted for library preparation. mRNA libraries were prepared by Stanford Genome Sequencing Center using TruSeq Stranded mRNA Library Preparation kit (Illumina). Four biological replicates representing independent cultures of *C. elegans* on independently prepared feeding RNAi were performed for each sample in this study.

RNA-seq and computational analysis mRNA libraries were pooled and paired-end sequenced for 100 bp, resulting in 40 million raw reads per sample. Raw reads were trimmed of adaptor sequences using fastx (http://hannonlab.cshl.edu/fastx_toolkit/) and mapped to *C. elegans* reference genome (*ce10*) using Tophat2 (*Kim et al., 2013*) and featureCounts (*Liao et al., 2014*). Uniquely mapped reads were used to calculate expression level of genes. Differential expression analysis was performed using DeSeq2 (*Love et al., 2014*). GO-term enrichment analysis of significantly upregulated or downregulated genes (FDR < 0.1) were performed through PANTHER gene ontology tool (*Thomas et al., 2003*). Raw sequencing data is accessible through NCBI GEO (Accession code: GSE125694). Full DESeq2 output and GO-term enrichment analysis results can be found on *Supplementary file 2*.

## Single molecule pulldown (SiMPull) experiments

SiMPull assays were performed as previously described (Zou et al., 2016). In brief, *C. elegans* grown on twenty 15 cm dishes were collected and washed, then dropped in liquid nitrogen to form 'worm pearls.' Worm pearls (50 mg wet weight) were thawed in 250 ul lysis buffer (50 mM HEPES pH 7.7, 50 mM KCl, 2 mM MgCl2, 250 mM Sucrose, 1 mM EDTA pH 8.0, with protease inhibitors). After brief sonication on ice (3' pulse with 30' pause, six cycles) to break the cuticle, 100 mM NaCl and 1% Triton X-100 were added into solution and samples were rotated at 4°C for 1 hr. After centrifugation at 16,000 g for 15 min, supernatants were transferred to new tubes and measured by BCA assay (Thermo Fisher Scientific) for total protein concentration. Worm lysates from different samples were adjusted to 7 mg/ml concentration by lysis buffer and used for SiMPull. Briefly, normalized lysates were incubated on quartz slides pre-coated with or without biotinylated anti-GFP antibodies (Rockland immunochemicals) to pull down ZTF-11::GFP, after washing away unbound sample, mCherry signals were recorded to visualize captured ZTF-11 binding partners. mCherry tagged proteins immobilized on the slides were visualized by a TIRF microscope equipped with excitation laser 561 nm, and DV2 dichroic 565dcxr dual-view emission filters (520/30 nm and 630/50 nm). Mean spot counts per image and standard deviation were calculated from images taken from 5 to 17 different regions.

## Acknowledgements

This work was supported by the Howard Hughes Medical Institute, NIH R37 NS048392, and NIH T32 GM007276.

Some strains were provided by the CGC, which is funded by NIH Office of Research Infrastructure Programs (P40 OD010440), and the MITANI Lab through the National Bio-Resource Project of the MEXT, Japan.

## Additional information

### Competing interests

Kang Shen: Reviewing Editor, eLife. The other authors declare that no competing interests exist.

### Funding

| Funder | Grant reference number | Author |
| --- | --- | --- |
| National Institutes of Health | T32 GM007276 | Joo Lee |
| Howard Hughes Medical Institute | | Joo Lee Caitlin A Taylor Kang Shen |
| National Institutes of Health | R37 NS048392 | Joo Lee Caitlin A Taylor Kang Shen |

The funders had no role in study design, data collection and interpretation, or the decision to submit the work for publication.

### Author contributions

Joo Lee, Conceptualization, Data curation, Formal analysis, Investigation, Visualization, Methodology, Writing—original draft, Writing—review and editing; Caitlin A Taylor, Data curation, Formal analysis, Investigation, Writing—review and editing; Kristopher M Barnes, Allison Chen, Data curation, Formal analysis; Ao Shen, Data curation, Formal analysis, Investigation; Emerson V Stewart, Conceptualization, Data curation, Supervision, Investigation; Yang K Xiang, Zhirong Bao, Supervision, Writing—review and editing; Kang Shen, Conceptualization, Supervision, Funding acquisition, Writing—review and editing

## Author ORCIDs

Joo Lee (iD) https://orcid.org/0000-0001-5875-6036
Caitlin A Taylor (iD) http://orcid.org/0000-0003-0016-9175
Zhirong Bao (iD) http://orcid.org/0000-0002-2201-2745
Kang Shen (iD) https://orcid.org/0000-0003-4059-8249

## Decision letter and Author response

Decision letter https://doi.org/10.7554/eLife.46703.048
Author response https://doi.org/10.7554/eLife.46703.049

## Additional files

### Supplementary files

• Supplementary file 1. Embryonic cell lineages expressing ZTF-11, up to 350-cell stage and their postmitotic identities.
DOI: https://doi.org/10.7554/eLife.46703.040

• Supplementary file 2. Differentially expressed genes in *ztf-11* knockdown embryos.
DOI: https://doi.org/10.7554/eLife.46703.041

• Supplementary file 3. Strains used for this study.
DOI: https://doi.org/10.7554/eLife.46703.042

• Supplementary file 4. DNA constructs and oligos used for this study.
DOI: https://doi.org/10.7554/eLife.46703.043

• Transparent reporting form
DOI: https://doi.org/10.7554/eLife.46703.044

### Data availability

Sequencing data have been deposited in GEO under accession code GSE125694. All data generated or analysed during this study are included in the manuscript and supporting files. Source data files have been provided whenever applicable.

The following dataset was generated:

| Author(s) | Year | Dataset title | Dataset URL | Database and Identifier |
|---|---|---|---|---|
| Lee J, Stewart EV, Taylor CA, Barnes KM, Chen A, Bao Z, Shen A, Shen K | 2019 | A Myt1 family transcription factor defines neuronal fate by repressing non-neuronal genes | https://www.ncbi.nlm.nih.gov/geo/query/acc.cgi?acc=GSE125694 | NCBI Gene Expression Omnibus, GSE125694 |

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
