## [Decision Letter]

Thank you for submitting your article "A Myt1 family transcription factor defines neuronal fate by repressing non-neuronal genes" for consideration by *eLife*. Your article has been reviewed by three peer reviewers, one of whom is a member of our Board of Reviewing Editors, and the evaluation has been overseen by Marianne Bronner as the Senior Editor. The following individual involved in review of your submission has agreed to reveal their identity: John Murray (Reviewer #2).

The reviewers have discussed the reviews with one another and the Reviewing Editor has drafted this decision to help you prepare a revised submission.

Summary:

This paper reports that a member of the Myt1 family of zinc finger transcription factors functions in *C. elegans* drives neurogenic fate while simultaneously blocking epidermal traits. This finding is exciting because Myt1 is routinely used to induce trans-differentiation of mammalian skin cells into neuroblasts. Remarkably, this work shows that ZTF-11, the sole Myt1 homolog in *C. elegans*, is also sufficient to transform epidermal precursors into neurons. Moreover, Lee et al. show that ZTF-11 drives an epidermal to neuronal trans-differentiation event that occurs during normal *C. elegans* development. These findings indicate that ZTF-11/Myt1 function is likely evolutionarily conserved and thus investigations that exploit the experimental tractability of *C. elegans* can effectively advance our understanding of the mechanism Myt1 function in mammals. The work deploys powerful experimental tools including a clever strategy for a conditional knockout of the ZTF-11/Myt1 locus that monitors an intrinsic GFP signal to detect cell-specific knockdown. In a first for any metazoan, the cell specific and temporal expression of ZTF-11 throughout embryonic development is defined by time-lapse imaging. Genetic methods also place *ztf-11* in a transcriptional cascade involving the proneural transcription factor genes *lin-22*/Hairy and *lin-32*/Achaete-Scute and demonstrate physical interaction with the conserved MuvB co-repressor complex.

Essential revisions:

Generally, the authors need to be commended for this very rigorous analysis. However, a number of loose ends were identified that the authors should tie up:

1) The ability of *ztf-11* to induce neuronal fates in V1-4 (in a *lin-32*-dependent manner) is a little unexpected given that *ztf-11* is a target of *lin-32* in V5. The authors need to test whether *ztf-11* and *lin-32* cross-regulate, i.e. (a) is the ability of *ztf-11* to induce neuronal fates in V1-4 accompanied by turning on *lin-32* expression and (b) in the V5 lineage, does *lin-32* also depend on *ztf-11* (feedback loop; in Figure 4A*ztf-11* expression appears to commence before *lin-32*, so the regulation of *ztf-11* by *lin-32* is perhaps the feedback). This is trivial thing to test and given the genetic *lin-32*-dependence such feedback almost has to exist.

2) An alternative explanation should be considered for results reported in Figure 6, in which pan neuronal marker gene is fully expressed in embryos homozygous for a *ztf-11* null allele. The authors conclude from this result that *ztf-11* is not necessary for differentiation of embryonic neurons. The problem with this interpretation is that it overlooks the more likely explanation of a maternal effect. The *ztf-11*(0) allele used for this experiment is lethal at the first larval stage and therefore must be maintained with a balancer chromosome. Thus, embryos that are homozygous for the *ztf-11* null allele will have maternal *ztf-11* mRNA that could be sufficient to sustain neuronal fate in the embryo but not for later stages that depend on zygotic expression. The authors should (a) use global *ztf-11* RNAi (see Figure 8) to knockdown maternal *ztf-11* mRNA in the balanced *ztf-11* strain; or alternatively, (b) in which the rescue the *ztf-11* null phenotype with a non-complex *ztf-11* array (e.g. fosmid), mark the array with a marker that labels P1 descendant (say *myo-3*), find animals with dorsal/ventral bwm loss (= P2 loss = loss in germline), then score their non-transgenic offspring (larvally arrested animals).

3) While the analysis of several postembryonic neuronal fates is very elegantly and well done, it is quite restricted. Plus, the analysis of neuronal identity in the embryo is not quite up to par to the quality of the analysis of postembryonic fates. The authors’ work hints at an extremely intriguing possibility that is mentioned by the authors but not very well carved out and broadly enough probed: is *ztf-11* function in neurogenesis restricted to those cases where the ectoblast from which the neuron is derived already is a somewhat differentiated cell? That's indeed obvious in the Y to PDA transdifferentiation event. But it's also the case of the V cells – which are also polarized differentiated skin cells which essentially "transdifferentiate" to generate things like the postdeirid. In such cases a gene like *ztf-11* is apparently required to "wipe out" the non-neuronal state. But is this really true for all postembryonically generated neurons that come from such a differentiated epidermal state? In contrast, in the embryo, neural precursors don't go through such a differentiated state – and indeed *ztf-11* does not appear to have much of a role in embryonic neurogenesis. The authors do of course recognize this, but could word this issue more crisply. For example, they say: "these results indicate that *ztf-11* is particularly important for neurons that are generated from an established epidermal lineage". "established" is not very clear – they should say that these are cells that have fully differentiated by a number of different criteria.

Anyway, that's potentially the most exciting point of the manuscript but the authors need to take a closer look at this potential dichotomy with a few more markers for embryonic and postembryonic neurons. Specifically:

Postembryonic: are really all neurons that are generated from differentiated ectoblasts affected? Currently the authors only look at V cell descendants and Y cells, using nifty Cre/Lox approaches. I suggest that they use the plain null to assess whether a few other lineages are affected as well. As markers, they could use specific transcription factor reporters that come on in the L1, right after the respective neuron is generated. Either the reporter will be off (i.e. case proven) or it's still on, but then the authors could look at the characteristic speckled morphology of neurons:

– T cell descendant PHC and Q cell descendants with *unc-86*(ot893) (a mNG tagged allele)

– K descendant DVB with *lim-6* fosmid integrant *wgIs387* (perhaps *oxIs12* will also work, see below)

– G1 descendant RMH with sem-2 fosmid reporter *otIs313*.

Embryonic (assuming that maternal rescue is not occurring, as discussed above): The authors state that the embryonically generated head neurons cannot be counted well (with a panneuronal marker in an L1-arrested animals). But actually, this can be done quite easily using the Fiji cell-counter plugins. Alternatively, and perhaps more clearly, to provide more insight on how extensive is the role of *ztf-11* on embryonically born neurons, the authors should look a little more systematcally in head neurons. I would simply suggest to use the 4 main neurotransmitter systems, assayed with *eat-4* (Glu), *cho-1* (ACh), *oxIs12* (GABA) and *cat-1* (aminergic). This covers most neurons. There's probably little need for precise cell ID here because the expectation is that there will be little if any change, i.e. simply a count of reporter-positive neuron in the head of L1-arrested *ztf-1* nulls should suffice.

This is all very simple marker analysis which will allow the authors to support the interesting case of a transcription factor that controls a specific type of neurogenic events.

---

## [Author Response]

Essential revisions:Generally, the authors need to be commended for this very rigorous analysis. However, a number of loose ends were identified that the authors should tie up:1) The ability of ztf-11 to induce neuronal fates in V1-4 (in a lin-32-dependent manner) is a little unexpected given that ztf-11 is a target of lin-32 in V5. The authors need to test whether ztf-11 and lin-32 cross-regulate, i.e. (a) is the ability of ztf-11 to induce neuronal fates in V1-4 accompanied by turning on lin-32 expression and (b) in the V5 lineage, does lin-32 also depend on ztf-11 (feedback loop; in Figure 4A ztf-11 expression appears to commence before lin-32, so the regulation of ztf-11 by lin-32 is perhaps the feedback). This is trivial thing to test and given the genetic lin-32-dependence such feedback almost has to exist.

To address the possibility of feedback loop between LIN-32 and ZTF-11, we have now investigated LIN-32 expression in ectopic neurons induced by ZTF-11 misexpression in V1-4. We found that misexpression of ZTF-11 can indeed turn on LIN-32 in those ectopic neurons (Figure 5—figure supplement 1). Since ectopic expression of *ztf-11* in V1-4 only induced ectopic neurons at a penetrance, we also examined the ZTF-11::GFP positive cells that do not show neuronal morphology and found that they did not turn on LIN-32 (Figure 5—figure supplement 1). Based on these results, we conclude that ZTF-11 can, in some contexts, induce expression of LIN-32 which leads generation of the ectopic neurons. As for the second point, we have already provided data suggesting that LIN-32 expression is independent on ZTF-11 where we show that the intensity of LIN-32 transcriptional reporter in V5 lineage is unchanged in ZTF-11 conditional knock-out (Figure 8F-G). In summary, these data suggest that the endogenous LIN-32 expression is likely independent of ZTF-11. However, ectopic expression of ZTF-11 might induce LIN-32 in specific context to induce neurogenesis.

2) An alternative explanation should be considered for results reported in Figure 6, in which pan neuronal marker gene is fully expressed in embryos homozygous for a ztf-11 null allele. The authors conclude from this result that ztf-11 is not necessary for differentiation of embryonic neurons. The problem with this interpretation is that it overlooks the more likely explanation of a maternal effect. The ztf-11(0) allele used for this experiment is lethal at the first larval stage and therefore must be maintained with a balancer chromosome. Thus, embryos that are homozygous for the ztf-11 null allele will have maternal ztf-11 mRNA that could be sufficient to sustain neuronal fate in the embryo but not for later stages that depend on zygotic expression. The authors should (a) use global ztf-11 RNAi (see Figure 8) to knockdown maternal ztf-11 mRNA in the balanced ztf-11 strain; or alternatively, (b) in which the rescue the ztf-11 null phenotype with a non-complex ztf-11 array (e.g. fosmid), mark the array with a marker that labels P1 descendant (say myo-3), find animals with dorsal/ventral bwm loss (= P2 loss = loss in germline), then score their non-transgenic offspring (larvally arrested animals).

To address the potential maternal contribution of ZTF-11 from the balanced mothers, we have further globally knocked-down *ztf-11* in the balanced mothers and scored their *ztf-11* null offspring. We used identical feeding RNAi clone that we have previously used to knock down ZTF-11::GFP during our RNA-seq experiments (~75% knockdown was observed with ZTF-11::GFP). We then counted the number of embryonic ventral cord motor neurons and found no significant changes compared to the *ztf-11* null mutants (from heterozygous mother) without feeding RNAi (Figure 7B). We have attempted the alternative approach suggested by the reviewer as well but could not generate a stable *ztf-11* rescue transgenic lines in the allocated time for revision experiments. Even so, we have noticed rare balancer-negative transgenic F1 animals which gave rise to arrested L1 offspring. These L1 worms still showed *rab-3*::GFP in many head neurons (no data shown). Based on these results, our data suggests that the lack of embryonic *ztf-11* phenotype is unlikely to be due to maternal contributions.

3) While the analysis of several postembryonic neuronal fates is very elegantly and well done, it is quite restricted. Plus, the analysis of neuronal identity in the embryo is not quite up to par to the quality of the analysis of postembryonic fates. The authors’ work hints at an extremely intriguing possibility that is mentioned by the authors but not very well carved out and broadly enough probed: is ztf-11 function in neurogenesis restricted to those cases where the ectoblast from which the neuron is derived already is a somewhat differentiated cell? That's indeed obvious in the Y to PDA transdifferentiation event. But it's also the case of the V cells – which are also polarized differentiated skin cells which essentially "transdifferentiate" to generate things like the postdeirid. In such cases a gene like ztf-11 is apparently required to "wipe out" the non-neuronal state. But is this really true for all postembryonically generated neurons that come from such a differentiated epidermal state? In contrast, in the embryo, neural precursors don't go through such a differentiated state – and indeed ztf-11 does not appear to have much of a role in embryonic neurogenesis. The authors do of course recognize this, but could word this issue more crisply. For example, they say: "these results indicate that ztf-11 is particularly important for neurons that are generated from an established epidermal lineage". "established" is not very clear – they should say that these are cells that have fully differentiated by a number of different criteria.Anyway, that's potentially the most exciting point of the manuscript but the authors need to take a closer look at this potential dichotomy with a few more markers for embryonic and postembryonic neurons. Specifically:Postembryonic: are really all neurons that are generated from differentiated ectoblasts affected? Currently the authors only look at V cell descendants and Y cells, using nifty Cre/Lox approaches. I suggest that they use the plain null to assess whether a few other lineages are affected as well. As markers, they could use specific transcription factor reporters that come on in the L1, right after the respective neuron is generated. Either the reporter will be off (i.e. case proven) or it's still on, but then the authors could look at the characteristic speckled morphology of neurons:– T cell descendant PHC and Q cell descendants with unc-86(ot893) (a mNG tagged allele)– K descendant DVB with lim-6 fosmid integrant wgIs387 (perhaps oxIs12 will also work, see below)– G1 descendant RMH with sem-2 fosmid reporter otIs313.Embryonic (assuming that maternal rescue is not occurring, as discussed above): The authors state that the embryonically generated head neurons cannot be counted well (with a panneuronal marker in an L1-arrested animals). But actually, this can be done quite easily using the Fiji cell-counter plugins. Alternatively, and perhaps more clearly, to provide more insight on how extensive is the role of ztf-11 on embryonically born neurons, the authors should look a little more systematically in head neurons. I would simply suggest to use the 4 main neurotransmitter systems, assayed with eat-4 (Glu), cho-1 (ACh), oxIs12 (GABA) and cat-1 (aminergic). This covers most neurons. There's probably little need for precise cell ID here because the expectation is that there will be little if any change, i.e. simply a count of reporter-positive neuron in the head of L1-arrested ztf-1 nulls should suffice.This is all very simple marker analysis which will allow the authors to support the interesting case of a transcription factor that controls a specific type of neurogenic events.

We thank the reviewer for these constructive suggestions. We crossed the *ztf-11* mutant into multiple markers, which label the postembryonic neuroectoblast lineages including Q, K, and G1. Animals were synchronized to early L1 by bleaching and then fed for 20 hours to allow development into late L1 stages. We then scored the expression of respective cell fate markers and found that postembryonic neurons born during L1, AVM/PVM, DVB, and RMH were indeed lost (Figure 6A-F) in *ztf-11* null. We did note that the number of UNC-86 expressing cells were reduced in the lumbar ganglion, potentially due to loss of T cell descendants (no data shown). Unfortunately, we found it challenging to reliably identify UNC-86-expressing postembryonic neurons (AQR/PQR, and T cell descendants) due to presence of nearby neurons and potential migration defects. In addition, we have looked into Pn lineage descendants which give rise to many ventral cord motor neurons (VMNs). We generated the *ztf-11* conditional KO (*ztf-11* cKO) mutant by crossing *dpy-7* epidermal Cre into floxed *ztf-11* allele. Interestingly, we found that the serotonergic VC4-5 neurons to be strongly affected by *ztf-11* cKO (*cat-1* staining lost in ~90% of animals). In cholinergic or GABAergic neurons, the phenotype was much weaker (*unc-17/unc-47* staining lost in ~20% and ~5% of animals respectively) (Figure 6G-H).

We have also systemically examined embryonic head neurons in *ztf-11* null mutants. We attempted to use an ImageJ plugin to automatically count confocal Z-stack images of a Pan neuronal marker in L1 animals. However, this method produced numbers that are dramatically fewer than expected in the wild type L1s, likely due to small size of early L1 animals. In the light of this technical difficulty, we performed manual counting of head neurons of 4 major neurotransmitter systems in wild type and *ztf-11* null animals. We could not find differences between wildtype and *ztf-11* null animals, which strengthened our previous conclusion that ZTF-11 was dispensable for embryonic neurogenesis (Figure 7C-F). Taken together, we have provided evidence that ZTF-11 is distinctively required for neurogenesis from multiple neuroectoblast lineages (we examined 6 out of 9 neuroectoblast lineages in worms) rather than embryonic neurogenesis. We have also worded this conclusion more visible in the manuscript as well.